# REINFORCEMENT LEARNING FOR ADAPTIVE MESH REFINEMENT

## ABSTRACT

Large-scale finite element simulations of complex physical systems governed by partial differential equations (PDE) crucially depend on adaptive mesh refinement (AMR) to allocate computational budget to regions where higher resolution is required. Existing scalable AMR methods make heuristic refinement decisions based on instantaneous error estimation and thus do not aim for long-term optimality over an entire simulation. We propose a novel formulation of AMR as a Markov decision process and apply deep reinforcement learning (RL) to train refinement *policies* directly from simulation. AMR poses a new problem for RL as both the state dimension and available action set changes at every step, which we solve by proposing new policy architectures with differing generality and inductive bias. The model sizes of these policy architectures are independent of the mesh size and hence can be deployed on larger simulations than those used at train time. We demonstrate in comprehensive experiments on static function estimation and time-dependent equations that RL policies can be trained on problems without using ground truth solutions, are competitive with a widely-used error estimator, and generalize to larger, more complex, and unseen test problems.

## 1 INTRODUCTION

Numerical simulation of PDEs via the finite element method (FEM) (Brenner & Scott, 2007) plays an integral role in computational science and engineering (Reddy & Gartling, 2010; Monk et al., 2003). Given a fixed set of basis functions, the resolution of the finite element mesh determines the trade-off between solution accuracy and computational cost. For complex systems with large variations in local solution characteristics, uniform meshes can be computationally inefficient due to their suboptimal distribution of mesh density, under-resolving regions with complex features such as discontinuities or large gradients and over-resolving regions with smoothly varying solutions. For systems with multi-scale properties in particular, attempting to resolve these features with uniform meshes can be challenging even on the largest supercomputers. To achieve more efficient numerical simulations, adaptive mesh refinement (AMR), a class of methods that dynamically adjust the mesh resolution during a simulation to maintain equidistribution of error, is used to significantly increase accuracy relative to computational cost.

Existing methods for AMR share the same iterative process of computing a solution on the current mesh, estimating refinement indicators, marking element(s) to refine, and generating a new mesh by refining marked elements (Bangerth & Rannacher, 2013; Červený et al., 2019). The optimal algorithms for error estimation and marking in many problems, especially evolutionary PDEs, are not known (Bohn & Feischl, 2021), and deriving them is difficult for complex refinement schemes such as $hp$-refinement (Zienkiewicz et al., 1989). As such, the current state-of-the-art is guided largely by heuristic principles that are derived by intuition and expert knowledge (Zienkiewicz & Zhu, 1992), but choosing the best combination of heuristics is complex and not well understood.

We advance the novel notion that adaptive mesh refinement is fundamentally a *sequential decision-making* problem in which a sequence of greedy decisions based on instantaneous error indicators does not constitute an optimal sequence of decisions for the actual goal of achieving high cumulative or terminal accuracy. In time-dependent problems for example, an error estimator by itself cannot preemptively refine elements which would encounter complex features in the next time step. This means that the optimality of a refinement decision depends on the accuracy of the future solution and that selecting an element which yields the largest reduction in error at the current time step may

Figure 1: AMR viewed as a Markov decision process.

not be the optimal decision over the entire simulation. Whether and how optimal AMR strategies can be found by directly optimizing a long-term performance objective are open questions.

Given this perspective, we formulate AMR as a Markov decision process (MDP) (Puterman, 2014) (Figure 1) and propose a reinforcement learning (RL) (Sutton & Barto, 2018) approach that explicitly trains a mesh refinement policy to optimize a performance metric, such as final solution error. In contrast to most, if not all, benchmark problems and complex applications of RL (Mnih et al., 2015; Brockman et al., 2016; Osband et al., 2019; Berner et al., 2019; Vinyals et al., 2019), AMR poses a new challenge as the sizes of both the state and the set of available actions depend on the current number of mesh elements, which changes with each refinement action at every MDP time step. While one may define a fixed and bounded state and action space given a finite refinement budget, doing so is very inefficient as the policy's input-output dimensions would have to accommodate the full exponentially large space but only subspaces (with increasing size) are encountered during simulation. In many practical applications, one would routinely encounter input dimensions on the order of millions or billions of degrees of freedom. This motivates the design of efficient policy architectures that leverage the correspondence between the current mesh state and valid action set.

In this paper, we make the following conceptual, methodological, and experimental contributions: 1) We formally define an MDP with effective variable-size state and action spaces for AMR (Section 3.2); 2) We propose three policy architectures—with differing generality, inductive bias, and capacity for modeling interaction—that operate on such variable-size spaces (Section 4); 3) As a path toward potentially solving large and complex problems on which RL cannot tractably be trained, we investigate the generalizability of policies trained on small representative features with known analytic solutions and the effectiveness of policies trained using a novel reward formulation that can be applied to problems without known analytic solutions (Section 5); 4) Our experiments demonstrate for the first time that RL can be competitive with, and sometimes outperform, a greedy refinement strategy based on the widely-used Zienkiewicz-Zhu-type error estimator; moreover, we show that an RL refinement policy can generalize to higher refinement budgets and larger meshes, transfer effectively from static to time-dependent problems, and can be effectively trained on more complex problems without readily-available ground truth solutions. (Section 6).

## 2 RELATED WORK

The formulation of problems in numerical analysis as statistical learning problems can be traced at least as far in time as to Poincaré (Poincaré, 1912; Diaconis, 1988). Contemporary works have employed neural networks as powerful function approximators in existing numerical PDE and linear system solvers to achieve faster convergence rates, generalize to different boundary conditions or larger problems, and approximate underresolved features in coarse-grained simulations (Hsieh et al., 2018; Luz et al., 2020; Bar-Sinai et al., 2019). Our work focuses on optimizing a finite element space rather than components of a numerical solver.

To the best of our knowledge, no prior work has formulated adaptive mesh refinement as a sequential decision-making problem and proposed a reinforcement learning approach (Sutton & Barto, 2018). Previous work at the intersection of neural networks and mesh-based simulation trained neural networks to predict mesh densities, sizes, or error fields for use by downstream mesh generators (Dyck et al., 1992; Chedid & Najjar, 1996; Zhang et al., 2020; Pfaff et al., 2020; Chen & Fidkowski, 2020). Brevis et al. (2020) apply supervised learning to find an optimal parameterized test space without modifying the degrees of freedom. Bohn & Feischl (2021) show theoretically that the estimation and marking steps of AMR for an elliptic PDE can be represented optimally by a recurrent neural network, but model optimization was left as an open question. Recent studies have leveraged the effectiveness of graph neural networks (GNN) (Sperduti & Starita, 1997; Gori et al., 2005; Scarselli et al., 2008) at representing relational structure to predict PDE dynamics on general unstructured and non-uniform meshes (Alet et al., 2019; Belbute-Peres et al., 2020; Pfaff et al., 2020). Previous

work on graph generation and formation have employed GNNs as the policy model in an RL context with applications to biological and social network datasets (You et al., 2018; Trivedi et al., 2020).

Learning a policy for unbounded variable-size state and action spaces is a rare—if not new—problem for RL, which has been typically applied to environments with fixed-size observation and small bounded action spaces in almost all benchmark problems (Mnih et al., 2015; Brockman et al., 2016; Osband et al., 2019). While there are notable applications where the available action set varies with state (Berner et al., 2019; Vinyals et al., 2019), they do not face the challenge of potentially millions of possible actions that arises in large-scale AMR. The technique of growing action spaces (Farquhar et al., 2020) maintains a fixed action space size within each episode, whereas both state and action space sizes change at every time step within an episode in AMR.

## 3 BACKGROUND AND FORMULATION

### 3.1 FINITE ELEMENT METHOD

Our mesh adaptation strategy is implemented in a FEM-based framework (Brenner & Scott, 2007). In FEM, the domain $\Omega \subset \mathbb{R}^D$ is modeled with a mesh that is a union of $E$ nonoverlapping subsets (*elements*) such that $\Omega := \bigcup \Omega_k$ where $k \in \mathbb{N} : k \leqslant E$. The solution on these elements is represented using polynomials (*basis functions*) which are used to transform the governing equations into a system of algebraic equations via the weak formulation. AMR is a commonly used approach to improve the trade-off between the solution accuracy, which depends on the shape and sizes of elements, and the computational cost, which depends on the number of elements. The most ubiquitous method for AMR is $h$-refinement, whereby elements are split into smaller elements (refinement) or multiple elements coalesce to form a single element (derefinement). In practical applications with unknown true solutions, the conventional AMR approach is to take greedy refinement decisions based on *a posteriori* error estimators, which rely on the numerical solution and its derived quantities on the current mesh, without regard to long-term optimality.

### 3.2 AMR AS A MARKOV DECISION PROCESS

We formulate AMR with spatial $h$-refinement[1] as a Markov decision process $\mathcal{M} := (\mathcal{O}, N_{\max}, \mathcal{A}, R, P, \gamma)$ with each component defined as follows. Each episode consists of $T$ RL time steps: for time-dependent PDEs, $T$ spans the entire simulation and there may be multiple underlying PDE evolution steps per RL step; for static problems, $T$ is an arbitrary number of steps at which RL can act. Consider a time step $t$ when the current mesh has $N_t \leq N_{\max} \in \mathbb{N}$ elements. Each element $i$ is associated with an *observation* $o_t^i \in \mathcal{O}$ and the *global state* is $s_t := [o_t^1, \ldots, o_t^N] \in \mathcal{O}^{N_t}$. We define $\mathcal{O} := \mathbb{R}^d$ such that each element's observation is a tensor of shape $d := l \times w \times c$ that includes the values and refinement depths of a local window centered on itself. For brevity, let $\mathcal{S}_t$ denote the current global state space $\mathcal{O}^{N_t}$. We denote an action by $a_t \in \mathcal{A}_t := \{0, 1, \ldots, N_t\} \subset \mathcal{A} := \{0, 1, \ldots, N_{\max}\}$, where 0 means "do-nothing" and $i \neq 0$ means refine element $i$. Given the current state and action, the MDP transition $P$ consists of:

1) refining the selected element into multiple finer elements (which increases $N_t$) if a refinement budget $B$ is not exceeded and the selected element is not at the maximum refinement depth $d_{\max}$;

2) stepping the finite element simulation forward in time (for time-dependent PDEs only);

3) computing a solution on the new finite element space.

When a true solution is available at *training* time, the reward at step $t$ is defined as the change in error from the previous step, normalized by the initial error to reduce variation across function classes:

$$r_t := (\|e_{t-1}\|_2 - \|e_t\|_2)/\|e_0\|_2 \,, \tag{1}$$

where error $e$ is computed relative to the true solution. With abuse of notation, we shall use $e$ to indicate the error norm. The ground truth is not needed to deploy a trained policy on test problems. When the true solution is not readily available, as is the case for most non-trivial PDEs, one may run a reference simulation on a highly-resolved mesh to compute equation 1, but this approach can be prohibitively expensive for training on large-scale simulations. Instead, we propose the use of a *surrogate reward* $r_t := \|u_{t,\text{refine}} - u_{t,\text{no-refine}}\|_2$, the normed difference between the estimated solution $u$ with and without executing the chosen refinement action. This surrogate, which is an upper bound

---

[1]Polynomial $p$-refinement can be formulated in a similar way. $r$-refinement (Huang & Russell, 2010; Dobrev et al., 2019) can be formulated as an RL problem but is not treated in this work.

on the true reward and effectively acts as an estimate of the error reduction, is only used at training time to minimize computational effort, whereas at test time, the effectiveness of trained policies is evaluated using the error computed with respect to a highly-resolved reference simulation.

Our objective to find a stochastic policy $\pi \colon \mathcal{S}_t \to \Delta(\mathcal{A}_t)$ to maximize the objective

$$J(\pi) := \mathbb{E}_{a \sim \pi(\cdot|s), s_{t+1} \sim P(\cdot|a, s_t)} \left[ \sum_{t=1}^{T} \gamma^t r_t \right] . \tag{2}$$

Aside from $\gamma \in (0, 1)$, this objective is equivalent to maximizing total error reduction: $e_0 - e_{\text{final}}$.

Although the size of the state vector and set of valid actions changes with each time step due to the varying $N_t$, this MDP is well-defined since one can define the global state space as the set of all possible $\mathcal{O}^N, N < N_{\text{max}}$, and likewise for the action space. Hence, the policy is navigating through subspaces of increasing size during an episode. Moreover, the exact 1:1 correspondence between the number of observation components and the number of valid actions calls for designing a dedicated policy architecture for AMR, which we present below in Section 4.

We work with the class of policy optimization methods as they naturally admit stochastic policies that could benefit AMR at test time: a stochastic refinement action could reveal the need for further refinement in a region that appears flat on a coarse mesh. We build on the policy gradient algorithm (Sutton et al., 2000; Schulman et al., 2017) to train a policy $\pi_\theta$ (parameterized by $\theta$) using batches of trajectories $\{\tau_b := \{(s_t, a_t, r_t)_k\}_{t=1}^{T}\}_{k=1}^{K}$ generated by the current policy.

## 4 POLICY ARCHITECTURES FOR VARIABLE STATE-ACTION SPACES

We propose three policy architectures, each with different inductive biases, that address the challenge of variable size state vector $s \in \mathbb{R}^{N_t \times d}$ and action set $\{0, 1, \ldots, N_t\}$, both of whose sizes changes with number of elements $N_t$ within an episode. These architectures are compatible with any stochastic policy gradient algorithm. We focus on the special case of 1:1 correspondence between the number of observations that compose each global state and the number of available actions at that state. Although not treated in this work, these policy architectures can be easily extended to the general case of 1:$k$ correspondence—e.g., $k = 2$ to include derefinement actions.

### 4.1 INDEPENDENT POLICY NETWORK

The Independent Policy Network (IPN) handles the 1:1 correspondence by mapping each observation to a probability for the corresponding action. Let $f_\theta \colon \mathbb{R}^d \mapsto \mathbb{R}$ be a function parameterized by $\theta$. Given a matrix of observations $s := [o^1, \ldots, o^N] \in \mathbb{R}^{N \times d}$, we define the policy as

$$\pi(\cdot|s) = \mathtt{softmax}\left( f_\theta(o^1), \ldots, f_\theta(o^N) \right) . \tag{3}$$

For example, using a neural network with hidden layer $\boldsymbol{W} \in \mathbb{R}^{d \times h}$ with $h$ nodes, output layer $\boldsymbol{H} \in \mathbb{R}^{h \times 1}$, and activation function $\sigma$, the discrete probability distribution over $N$ actions conditioned on $s$ is defined by $\mathtt{softmax}\left( \sigma(s\boldsymbol{W})\boldsymbol{H} \right)$.

IPN applies to meshes of any size since the set of trainable parameters $\theta$ is independent of $N$, but it has two main limitations. Firstly, it makes a strong assumption of locality as the action probability at an element does not depend on the observations at other elements. This assumption also appears in existing AMR methods that estimate error independently at each element; in fact, the output probabilities of IPN may be viewed as normalized error estimates. Secondly, the permutation equivariance of this architecture—i.e., $\pi(a^{\mu(i)}|(o^{\mu(1)}, \ldots, o^{\mu(N)})) = \pi(a^i|s)$ for any permutation operator $\mu \colon [N] \mapsto [N]$—means that one cannot use the ordering of inputs to represent spatial relations among elements, which would be necessary for refining an element based on neighboring conditions. We mitigate this problem by defining each element's observation as an image tensor that includes neighborhood information and using a convolutional network layer, but this may face difficulties on unstructured meshes with non-quadrilateral elements (Červený et al., 2019).

### 4.2 HYPERNETWORK POLICY

The hypernetwork policy captures higher-order interaction among inputs via the function form

$$\pi(\cdot|s) = \mathtt{softmax}\left( f_{g_\phi(s)}(o^1), \ldots, f_{g_\phi(s)}(o^N) \right) . \tag{4}$$

The main policy network weights $\theta$ are now the output of a hypernetwork (Ha et al., 2017) $g_\phi \colon \mathbb{R}^{N \times d} \mapsto \mathbb{R}^{\dim(\theta)}$, parameterized by $\phi$, which produces mixing among the inputs $s \in \mathbb{R}^{N \times d}$. Continuing with the example in IPN, where the policy network's first layer is $\boldsymbol{W} \in \mathbb{R}^{d \times h}$, a hypernetwork with two layers can be instantiated as $\left[ \sum_{i=1}^{N} (s \boldsymbol{U})_{i,:} \right] \boldsymbol{V} = \boldsymbol{W}$ where $\boldsymbol{U} \in \mathbb{R}^{d \times h_1}$ and $\boldsymbol{V} \in \mathbb{R}^{h_1 \times (d \times h)}$ are the trainable parameters $\phi$, and $\boldsymbol{M}_{i,:}$ denotes the $i$-th row of matrix $\boldsymbol{M}$. The output $\boldsymbol{W}$ can then be used as part of $\theta$ in equation 3.

This increased generality comes with more difficulty in the choice of architecture (specific form of $g_\phi$), which affects the extent to which it captures interaction among inputs. It does not contain an inductive bias for the local nature of interactions seen in classical applications of AMR. In fact, the use of a summation from $i = 1$ to $N$ in the example above means that complete global information affects each local refinement decision, which is an extremely strong inductive bias.

### 4.3 GRAPH NETWORK POLICY

We build on graph networks (Scarselli et al., 2008; Battaglia et al., 2018) to address both the issue of interaction terms and spatial relation among elements. Specifically, we construct a policy based on Interaction Networks (Battaglia et al., 2016), which is a special case without global attributes[2]. At each step, the mesh is represented as a graph $\mathcal{G} = (V, E)$. Each vertex $v^i$ in $V = \{v^i\}_{i=1:N}$ corresponds to element $i$ and is initialized to be the observation $o^i$. $E = \{(e^k, r^k, s^k)\}_{k=1:N^e}$ is a set of edges with attributes $e^k$ between sender vertex $s^k$ and receiver vertex $r^k$. An edge exists between two vertices if and only if they are spatially adjacent. We define the initial edge attribute $e^k$ as a one-hot vector indicator of the difference in refinement depth between $r^k$ and $s^k$.

Graph networks capture the relations between nodes and edges via the inductive bias of its internal update rules. A single *forward pass* through the graph policy involves one or many rounds of *message passing* (Algorithm 1 in the Appendix). Each round is defined by the following sequence of computations: 1) Each edge attribute $e^k$ is updated by learned function $\varphi^e$ using local node information via $\hat{e}^k \leftarrow \varphi^e(e^k, v^{r^k}, v^{s^k})$; 2) For each node $i$, we denote by $\hat{E}^i := \{(\hat{e}^k, r^k, s^k)\}_{r^k = i}$ the set of all edges with node $i$ as the receiver, and all updated edge attributes are aggregated into a single feature $\bar{e}^i \leftarrow \rho^{e \rightarrow v}(\hat{E}^i)$ by aggregation function $\rho^{e \rightarrow v}$ (e.g., element-wise sum); 3) Then, each node attribute is updated by $\hat{v}^i \leftarrow \varphi^v(\bar{e}^i, v^i)$ using learned function $\varphi^v$. Each additional round increases the size of the local neighborhood that determines node attributes. Finally, we map each node attribute to a scalar using learned function $\psi$, apply a global softmax over all nodes, and interpret the value at each node $i$ as the probability of choosing element $i$ for refinement.

These update rules allow the graphnet policy to address both limitations of the IPN and the hypernetwork policy. Cross terms arise in the forward pass due to mutual updates of edge and node attributes using local information. The order of cross terms increases with each message-passing round. Local spatial relations between mesh elements are included by construction in the initial edge attributes, so there is no need to include numerical spatial information in each element's observation vector.

## 5 EXPERIMENTAL SETUP

Our experiments assess the ability of RL, using the proposed policy architectures, to find AMR strategies that generalize to test function classes that differ from the training class, generalize to variable mesh sizes and refinement budgets, and extend to more complex problems without readily-available ground truth solutions. We define the FEM environment in Section 5.1, the train-test procedure in Section 5.2, and the implementation of our method and baselines in Section 5.3.

### 5.1 AMR ENVIRONMENT

**MFEM.** We use MFEM (Anderson et al., 2021; MFEM), a modular open-source C++ library for FEM, to implement the MDP for AMR. We ran experiments on two classes of AMR problems: static and time-dependent. In the static case, the objective of mesh refinement is to minimize the $L^2$ error norm of projecting a variety of test functions onto a two-dimensional finite element space. In the time-dependent case, the functions are projected onto the finite element space, and a PDE of the form $\frac{\partial u}{\partial t} + \nabla \cdot \mathbf{F}(u) = 0$ is solved on a periodic domain using the finite element framework.

---

[2]While not demonstrated in this work, it is conceivable to use function coefficients and initial/boundary conditions as global attributes to improve generalization in a graphnet policy for AMR.

In contrast to the static case, the numerical error accumulated at each time step propagates with the physical dynamics and determines future error. Two types of PDEs were used: the linear advection equation, where $\mathbf{F}(u) = \mathbf{c}u$ with $\mathbf{c} = [1, 0]$, and the nonlinear Burgers equation, where $\mathbf{F}(u) = \mathbf{c}u^2$ with $\mathbf{c} = [1, 0.3]$. The advection equation is used as there is an analytic solution to provide a ground truth, whereas the Burgers equation is used as the resulting solutions are representative of more complex phys-

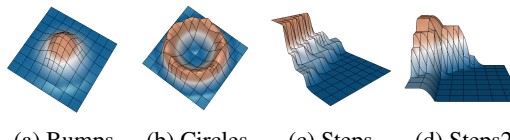

(a) Bumps    (b) Circles    (c) Steps    (d) Steps2

Figure 2: Individual samples from each true solution function class. Each function sampled in *bumps* and *circles* is a superposition of a random number of such features in general. Refinements shown here are produced by IPN.

ical systems that include shock and rarefaction waves, albeit without a readily-available analytic solution. The solution is represented using continuous (or discontinuous) second-order Bernstein polynomials for the static (or time-dependent) case, and the initial mesh is partitioned into $n_x \times n_y$ quadrilateral elements.

**True solutions.** We aim to show that RL policies have the potential to be deployed on larger and more complex problems without *a priori* known solutions by either training these policies on small representative features with known solutions or utilizing the proposed surrogate reward at training time. As such, we defined a collection of parameterized function classes, each exhibiting features such as sharp discontinuities and smooth variations, from which we randomly sample ground truth functions $f \colon [0, 1]^2 \mapsto \mathbb{R}$ to initialize each episode. The collection, shown in Figure 2 and defined precisely in Appendix A.1.2, includes: *bumps*, *circles*, *steps*, and *steps2* (a combination of two steps). These functions with closed form allow us to compute the error and reward at train time for static and advection problems. In the case of Burgers equation where the exact solution is not readily-available, we either use reference simulations on a highly-resolved mesh to act as a "ground truth" or employ the surrogate reward (defined in Section 3.2) to compute the reward for training.

In the static case, the true solution is fixed and each simulation time step is an RL step. For the time-dependent PDE cases, the initial solution is transported through the periodic domain and the ratio of simulation time steps to RL steps is set such that a feature advecting at unit velocity returns to its original position after 10 RL steps. We set refinement budget $B = 10$ for static problems, $B = 20$ for advection, and $B = 50$ for Burgers; episode length at train time equals $B$. Due to the Gibbs phenomena in FEM, using smooth polynomial approximations to solve hyperbolic systems containing discontinuities can introduce spurious oscillations which, in turn, can cause the simulation to become unstable. Therefore, we limit the true solutions to smooth functions (e.g., *bumps*, *circles*) for the advection and Burgers cases. Due to the nonlinearity of Burgers equation, initially smooth solutions can develop discontinuities in finite time. This behavior is resolved using the flux-corrected transport (FCT) approach (Boris & Book, 1997).

## 5.2 EXPERIMENTS AND PERFORMANCE METRIC

We conducted the following experiments to compare RL policies with baselines:

**In-distribution**: Train and test on true solutions sampled from the same function class.

**Out-of-distribution:** For problems such as Burgers equation where training on multiple initial conditions (ICs) may be expensive, we show the effectiveness of policies trained on a single initial condition (IC) when tested on multiple random ICs, either with or without fine-tuning.

**Generalization**: 1) **Static→advection**: Policies trained on static functions are tested on advection. 2) **Budget↑**: Policies trained with a small refinement budget $B$ (20 on static and 10 on advection) are test with $B = 50, 100$. 3) **Size↑**: Policies trained on an $8 \times 8$ initial mesh are tested on $16 \times 16$ and $64 \times 64$ initial meshes with and without preserving the relative solution and mesh length scales.

We define the performance of a given refinement policy in an episode in the static case as $(e_{\text{initial}} - e_{\text{final}})/e_{\text{initial}}$, where $e_{\text{initial}}$ (or $e_{\text{final}}$) is the error norm at the beginning (or end) of an episode, to remove the variation in the error due to different true solution classes and random function initialization within each class. In the time-dependent case, without any refinement, the error may increase over the course of the simulation due to the accumulation of discretization error. Hence, given a refinement policy that achieves $e_{\text{final}}$ at episode termination, we define its performance as $(e_{\text{no-refine, final}} - e_{\text{final}})/e_{\text{initial}}$, where $e_{\text{no-refine, final}}$ is the final error without any refinement.

We tuned hyperparameters by training on a multitask scenario for static and advection, and individually for Burgers; the procedure and chosen values are given in Appendix A.3. For every experiment and every policy architecture, we trained four independent policies with different random seeds. For each test case, we report the mean and standard error—over the four independent policies and with different simulator seeds —of the mean performance metric over 100 test episodes. At each test episode, we ensured that all methods faced the same initial condition (which differs across episodes).

### 5.3 Implementation and baselines

We describe the high-level implementation here and provide complete details in Appendix A.2. All policy architectures use a convolutional neural network with the same architecture as the input layer. The **IPN** has two fully-connected hidden layers with $h_1$ and $h_2$ nodes and `ReLU` activation, followed by a `softmax` output layer. Its action on input states is described in Section 4.1. The **Graphnet** policy is implemented with the Graph Nets library (Battaglia et al., 2018). Each input state consists of node observation tensors, all edge vectors, and the adjacency matrix. Node tensors are first passed through an Independent block, after which multiple Interaction networks (Battaglia et al., 2016) act on both node and edge embeddings to produce a probability at each node (see Section 4.3). The **Hypernet policy** is parameterized by matrices $U \in \mathbb{R}^{d \times h_1}$, $V \in \mathbb{R}^{h_1 \times (d \times h)}$, and $Y \in \mathbb{R}^{d \times h}$, where $h_1$ and $h$ are design choices. $U$ and $V$ act on input state $s$ to produce the main policy weights $W \in \mathbb{R}^{d \times h}$ (see Section 4.2) while $Y$ acts on $s$ to produce a bias $b \in \mathbb{R}^h$, so that the main policy's first hidden layer is $\text{ReLU}(sW + b)$. Output probabilities are computed in the same way as IPN.

**Baselines.** The **ZZ** policy uses a Zienkiewicz-Zhu-type recovery-based error estimator (Zienkiewicz & Zhu, 1992) and refines the element with the largest estimated error. The **TrueError** policy refines the element where the error of the numerical solution with respect to the true solution is largest. It is not the theoretical upper bound on performance because refining the element with largest error does not necessarily result in largest reduction of error but it does effectively pose an upper bound on the performance of an instantaneous error estimator. The **GreedyOptimal** policy performs one-step lookahead by checking all possible outcomes of refining each element individually and chooses the element whose refinement would result in the lowest error at the next step. In comparison to RL policies, only the ZZ policy can feasibly be deployed at test time as TrueError cannot be deployed on systems without known solutions and GreedyOptimal is intractable for real applications.

## 6 Results

We find that the proposed methods achieve performance that is competitive with baselines and, more importantly, generalize well to larger refinement budgets and mesh sizes, and transfer effectively from a static problem to a time-dependent problem. Videos of policies on advection and Burgers can be viewed at `https://sites.google.com/view/iclr2022-amr`.

### 6.1 In-distribution

**Static functions** (Figure 3a). RL policies either meet or significantly exceed the performance of ZZ on all function classes. Notably, both IPN and Graphnet outperform ZZ significantly on *steps* by spending the limited refinement budget only on regions with discontinuities (Figure 2c). On the smoother function classes such as *bumps* where ZZ is known to perform well, all three policy architectures have comparable performance to ZZ. Overall, IPN outperforms both Graphnet and Hypernet, while Hypernet performed poorly (albeit still better than random) on all classes except

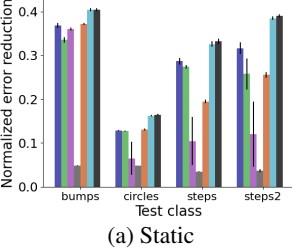
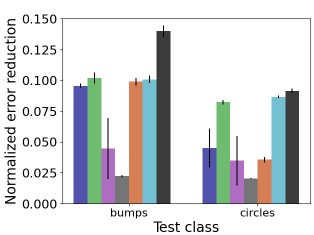
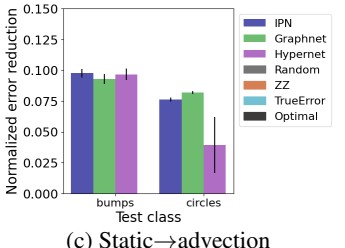

| (a) Static | (b) Advection | (c) Static→advection |

**Figure 3: In-distribution** and **Static→advection**. Performance of IPN, Graphnet and Hypernetwork policies versus baselines. Higher values are better. (a,b) RL policies were trained and tested on the same function class, for static and advection cases independently. (c) Static-trained policies on a function class are tested on advection of the same function class.

*bumps*. This suggests that capturing higher-order interaction among observations, each of which already contains local neighborhood information, is unnecessary for estimation of static functions as they only have a local domain of influence. Hypernetwork policies converged to the behavior of making no refinements on at least one out of four independent runs on all classes except *bumps*. This could be attributed to the inherent difficulty of choosing and training a highly nonlinear model.

**Advection** (Figure 3b). As explained above, we limit the true solutions to smooth functions (*bumps* and *circles*) in the advection case. Graphnet significantly outperformed ZZ on *circles* and is comparable to TrueError on *bumps*, while IPN is comparable to ZZ on both functions. Hypernet is comparable to ZZ on *circles* but has high variance across independent runs. Graphnet's higher performance than other methods indicates that its inductive bias can better represent the local geometric relations between neighboring mesh elements along the circle.

**Burgers equation**. In experiments with a single bump function (visualized in Figure 4c) as the fixed initial condition (IC) in both train and test, IPN trained with both the exact and surrogate rewards outperformed all baselines (Figure 4a). The policy trained using the surrogate reward resulted in similar performance to the policy trained with the exact reward, indicating that the surrogate reward can be effectively used to train policies without the need for a ground truth solution, as is necessary for general random ICs.

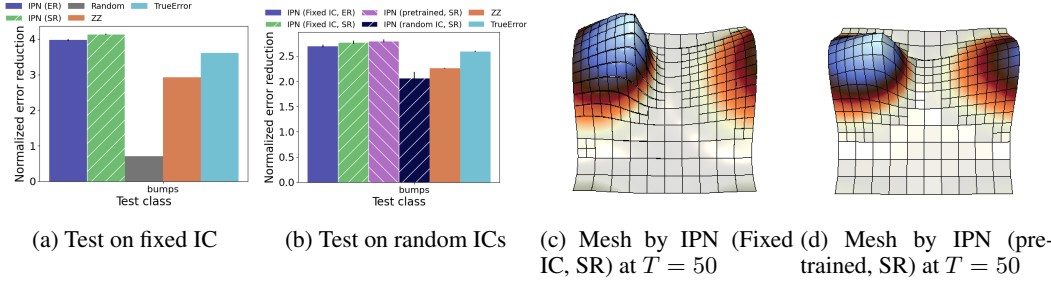

(a) Test on fixed IC  (b) Test on random ICs  (c) Mesh by IPN (Fixed  (d) Mesh by IPN (pre-
IC, SR) at $T = 50$  trained, SR) at $T = 50$

Figure 4: **Burgers equation** and **surrogate reward**. Solid bars/ER denote exact reward, striped bars/SR denote surrogate reward. (a) IPN trained and *tested on a fixed IC*. (b) IPN *tested on random ICs* using policies trained on a fixed IC (from Figure 4a), policies pretrained on fixed IC and fine-tuned on random ICs, and policies only trained on random ICs. (c/d) Visualization of resulting meshes for Burgers equation with a fixed bump IC.

## 6.2 OUT-OF-DISTRIBUTION (OOD)

Figure 4b shows the performance of IPN RL policies and baselines on Burgers equation with random ICs. We observe that policies trained on a fixed IC using either the exact or surrogate reward (labeled "Fixed IC, ER" and "Fixed IC, SR") generalize well to random unseen ICs and still outperform baselines. Moreover, policies that were pretrained with the surrogate reward on the fixed IC for 2k episodes and fine-tuned with the surrogate reward on random ICs for another 2k episodes performed the best ("pretrained, SR"). We truncated training around 4-5k episodes due to computational limits, but we see that performance can improve with more training episodes. Policies trained only on random ICs with the surrogate reward were not as performant, indicating that the training time was not sufficient and that pretraining on a fixed IC is a more efficient approach. We do not envision that one would train and test on highly different function classes in applications involving static and advection problems, but we include experimental results of all combinations of train and test function classes in Appendix B.

## 6.3 GENERALIZATION

**Static→advection** (Figure 3c). All static-trained policies demonstrated comparable performance to ZZ and TrueError when tested on *advection-bumps*, while both IPN and Graphnet significantly outperformed ZZ on *advection-circles*. Surprisingly, static-trained IPN significantly outperforms advection-trained IPN when tested on *advection-circles*, and the static-trained Hypernet does so as well on *advection-bumps*, while static-trained Graphnet maintains comparable performance to its advection-trained counterpart (Figure 3b vs. Figure 3c). Figure 12 shows that a static-trained policy on *bumps* with $B = 10$ correctly refines the region of propagation on *advection-bumps* with $B = 50$.

**Budget**↑ (Figure 5). RL policies trained with low refinement budget generalize to test cases with higher budget. In the static case, comparing Figure 3a ($B = 10$) with Figure 5a ($B = 50$) shows that

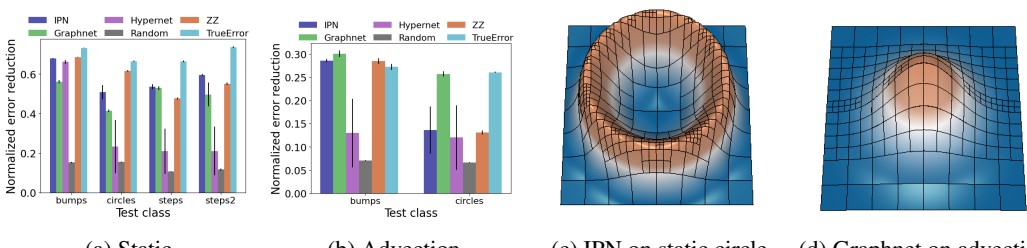

| (a) Static | (b) Advection | (c) IPN on static circle | (d) Graphnet on advection |

Figure 5: **Budget**↑. (a-b) Policies trained with budget $B = 10$ (static) and $B = 20$ (advection) are tested with $B = 50$. (c) IPN trained with $B = 10$ generalizes to $B = 100$. (d) Graphnet trained on advecting bump with $B = 20$ generalizes to $B = 50$.

the performance of RL policies relative to ZZ is generally preserved by the increase in refinement budget. Figures 5 and 8 show that an IPN trained with $B = 10$ makes qualitatively correct refinement decisions when allowed $B = 100$ during test. In the advection case (Figure 5b), Graphnet trained with $B = 20$ significantly outperforms both ZZ and TrueError when tested with $B = 50$ on *bumps* and comes within the margin of error of TrueError on *circles*. Figure 7 shows that an IPN trained with $B = 20$ correctly allocates a higher budget $B = 100$ to the limited region of propagation.

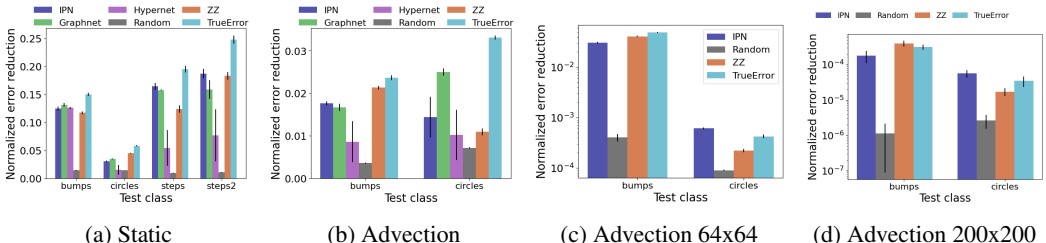

| (a) Static | (b) Advection | (c) Advection 64x64 | (d) Advection 200x200 |

Figure 6: **Size**↑. (a-b) Policies trained with initial $8 \times 8$ mesh were tested on initial $16 \times 16$ mesh. (c-d) Policies trained on $8 \times 8$ initial mesh were tested on initial meshes of size $64 \times 64$ (c) and $200 \times 200$ (d) with approximately constant feature-to-mesh length scale ratio.

**Size**↑ (Figure 6). In the static case, the relative performance of RL policies that were trained with an $8 \times 8$ initial mesh (Figure 3a) is generally preserved when deployed on a $16 \times 16$ initial mesh (Figure 6a). All policy architectures outperform ZZ on *bumps*, while IPN and Graphnet still outperform ZZ on *steps*. IPN and Graphnet were comparable to ZZ on $8 \times 8$ but underperformed on $16 \times 16$ on *circles*. Nonetheless, Figure 10 shows that IPN makes qualitatively correct refinements. On advection, relative performance is preserved on *circles* while IPN and Graphnet deproved slightly on *bumps* (Figure 3b vs. Figure 6b). Without preserving the solution-to-mesh length scales, the prior tests emulate deploying a policy trained on coarser versions of the simulations. When tested on $64 \times 64$ and $200 \times 200$ initial meshes that preserves the relative solution-to-mesh length scales, which emulates deploying a policy trained on a small subset of a highly-resolved simulation, Figures 6c and 6d shows that IPN is competitive with baselines on bumps and even outperforms TrueError on circles.

## 7   CONCLUSION

We present a novel formulation of adaptive mesh refinement as a Markov decision process and propose new policy architectures for scalable application of reinforcement learning. Our experiments on static and time-dependent problems demonstrate that RL policies can outperform a policy based on the widely-used ZZ-type error estimator, and in some cases even outperform a policy based on the exact true error, suggesting that RL can potentially provide efficiency gains beyond the reach of existing AMR approaches. We demonstrated that these RL policies generalize to different refinement budgets and larger meshes, transfer from static to time-dependent settings, and generalize to more complex problems even when trained without ground truth rewards. Moreover, because these RL policies do not use problem-specific knowledge or domain expertise as input, our results provide a path for learning novel AMR strategies for cases such as $hp$-refinement that currently lack effective solutions. Future work can extend our RL methods to include derefinement actions, sampling multiple elements at each time step, and taking a multi-agent perspective that views each element as an agent who acts concurrently with all other agents.

REPRODUCIBILITY STATEMENT

We have described the experimental protocol involving tuning, training, and testing in Section 5.2; the high-level implementation of the proposed method in Section 5.3; the specific implementation details of the proposed method in Appendix A.2; the experimental test cases in Appendix A.1.2 and Table 1; the detailed tuning procedure in Appendix A.3; and the complete specification of all hyperparameters in Appendix A.3, Table 2, and Table 3.

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

---

**Algorithm 1** Graphnet policy forward pass

---

1: **for** each message-passing round **do**
2:     **for** $k \in \{1, \ldots, N^e\}$ **do**
3:         $\hat{e}^k \leftarrow \varphi^e(e^k, v^{r^k}, v^{s^k})$ # Update edge attribute
4:     **end for**
5:     **for** $i \in \{1, \ldots, N\}$ **do**
6:         $\hat{E}^i := \{(\hat{e}^k, r^k, s^k)\}_{r^k = i}$ # Edge set for $v^i$
7:         $\bar{e}^i \leftarrow \rho^{e \to v}(\hat{E}^i)$ # Aggregation for $v^i$
8:         $\hat{v}^i \leftarrow \varphi^v(\bar{e}^i, v^i)$ # Update vertex attribute
9:     **end for**
10:   $e^k \leftarrow \hat{e}^k, \forall k \in [N^e], v^i \leftarrow \hat{v}^i, \forall i \in [N]$
11: **end for**
12: $\mathbb{R} \ni x^i \leftarrow \psi(v^i), \forall i \in [N]$
13: $\pi(a^i | s)$ is the $i$-th entry of softmax$(x^1, \ldots, x^N)$

---

## A   EXPERIMENTAL SETUP

### A.1   ENVIRONMENT DETAILS

#### A.1.1   MFEMCTRL

To interface between the MFEM framework and the RL environment, we developed MFEMCtrl, a C++/Python wrapper for the AMR and FEM capabilities in MFEM. MFEMCtrl is used to convert solutions to observations, apply refinement decisions, and calculate errors.

The initial mesh is partitioned into $n_x \times n_y = 8 \times 8$ elements for static and advection experiments and $10 \times 10$ for Burgers equation. Generalization experiments on larger initial mesh used $n_x \times n_y = 16 \times 16$ or $64 \times 64$. The true solution is projected onto the finite element space by interpolation to the nodes of the Bernstein basis functions. After each refinement action, the solution is projected again onto the refined mesh (for the static case) or integrated in time until the next refinement action (for the time-dependent case). The maximum refinement depth is fixed by the parameter $d_{\max}$ such that the maximally-refined mesh consists of $2^{d_{\max}} n_x \times 2^{d_{\max}} n_y$ elements. $d_{\max}$ was set to 3 for static experiments, whereas $d_{\max} = 2$ for advection and $d_{\max} = 1$ for Burgers equation due to the time step restrictions imposed by the Courant-Friedrichs-Lewy (CFL) condition of the finest elements.

**Observation.** The observation consisted of the solution and the depth of each element. Since the gradients of the solution are, by definition, a function of the solution, the observation does not include the gradients as they can be implicitly learned. The solution/depth of each element was observed by interpolating the functions to a local equispaced mesh (*image*) centered around each element, shown by the white box in Figure 1. Each element's observation is a $l \times w \times c$ tensor where $l = w = l_{\text{element}} + 2l_{\text{context}}$ is the spatial observation window with $l_{\text{element}} = 16$ sampled points inside the element and $l_{\text{context}} = 4$ sampled points in a coordinate direction outside the element. We chose $c = 2$ channels so that estimated function values and element depths are observed, while gradients are omitted since the policy network can in principle estimate gradients from the value channel. To impose a 1:1 map between each observation and possible action, we append a dummy $o^0$ to state $s$ corresponding to action 0. At most one refinement is allowed per MDP step.

Table 1: Parameterized true solutions

| | Parameter | [min, max] |
|---|---|---|
| Bumps (static) | $c_x$ | [0.2, 0.9] |
| | $c_y$ | [0.2, 0.9] |
| | $w$ | [0.05, 0.2] |
| | $n$ | $\{1, \ldots, 6\}$ |
| Bumps (advection) | $c_x$ | [0.3, 0.7] |
| | $c_y$ | [0.3, 0.7] |
| | $w$ | [0.005, 0.05] |
| | $n$ | $\{1, \ldots, 4\}$ |
| Bumps (Burgers) single IC | $c_x$ | 0.5 |
| | $c_y$ | 0.5 |
| | $w$ | 0.05 |
| | $n$ | 1 |
| Bumps (Burgers) random IC | $c_x$ | [0.3, 0.7] |
| | $c_y$ | [0.3, 0.7] |
| | $w$ | [0.005, 0.05] |
| | $n$ | $\{1, \ldots, 4\}$ |
| Circles (static) | $c_x$ | [0.2, 0.8] |
| | $c_y$ | [0.2, 0.8] |
| | $r$ | [0.05, 0.2] |
| | $w$ | [0.1, 1.0] |
| | $n$ | $\{1, \ldots, 6\}$ |
| Circles (advection) | $c_x$ | [0.3, 0.7] |
| | $c_y$ | [0.3, 0.7] |
| | $r$ | [0.05, 0.2] |
| | $w$ | [0.03, 0.05] |
| | $n$ | $\{1, \ldots, 4\}$ |
| Steps and Steps2 | $o$ | [0, 1.0] |
| | $\theta$ | $[0, \pi/2]$ |
| | $n$ | $\{1, \ldots, 6\}$ |

Steps

$$n \sim \text{Uniform}[n_{\min}, n_{\max}]$$
$$\theta \sim \text{Uniform}[\theta_{\min}, \theta_{\max}]$$
$$o_i \sim \text{Uniform}[o_{\min}, o_{\max}], \quad i = 1, \ldots, n$$
$$f(x, y) = \sum_{i=1}^{n} 1 + \tanh\left[100(o_i - (x + y \tan \theta))\right]$$

Steps2

$$n \sim \text{Uniform}[n_{\min}, n_{\max}]$$
$$\theta_i \sim \text{Uniform}[\theta_{\min}, \theta_{\max}], \quad i = 1, \ldots, n$$
$$o_i \sim \text{Uniform}[o_{\min}, o_{\max}], \quad i = 1, \ldots, n$$
$$s_i := (x - 0.5) \cos \theta_i - (y - 0.5) \cos \theta_i$$
$$f(x, y) = \frac{1}{2} \sum_{i=1}^{n} 1 + \tanh\left[100(s_i - o_i)\right]$$

### A.1.2 GROUND TRUTH FUNCTIONS

Bumps

$$n \sim \text{Uniform}[n_{\min}, n_{\max}]$$
$$c_{x,i} \sim \text{Uniform}[c_{x,\min}, c_{x,\max}], \quad i = 1, \ldots, n$$
$$c_{y,i} \sim \text{Uniform}[c_{y,\min}, c_{y,\max}], \quad i = 1, \ldots, n$$
$$w_i \sim \text{Uniform}[w_{\min}, w_{\max}], \quad i = 1, \ldots, n$$
$$f(x, y) = \sum_{i=1}^{n} \exp\left(-\frac{(x - c_{x,i})^2 + (y - c_{y,i})^2}{w_i}\right)$$

Circles

$$n \sim \text{Uniform}[n_{\min}, n_{\max}]$$
$$c_{x,i}, c_{y,i} \sim \text{Uniform}[c_{\min}, c_{\max}], \quad i = 1, \ldots, n$$
$$r_i \sim \text{Uniform}[r_{\min}, r_{\max}], \quad i = 1, \ldots, n$$
$$w_i \sim \text{Uniform}[w_{\min}, w_{\max}], \quad i = 1, \ldots, n$$
$$f(x, y) = \sum_{i=1}^{n} \exp\left(-\frac{(\sqrt{(x - c_{x,i})^2 + (y - c_{y,i})^2} - r_i)^2}{w_i}\right)$$

### A.1.3 REWARD

Let $u_t$ denote the true solution at time $t$, let $\hat{u}_t$ denote the estimated solution on the mesh at time $t$. For a given mesh at time $t-1$, a given time evolution of the true solution from $t-1$ to $t$, and a refinement action $a_t$ (which may be "do-nothing"), let $\hat{u}_{t,\text{refine}}$ denote the estimated solution on the mesh at time $t$ that has undergone refinement action $a_t$, and let $\hat{u}_{t,\text{no-refine}}$ denote the estimated solution on the mesh at time $t$ without that refinement. We have two reward definitions:

1. Delta norm reward with true solution

$$r_t := (e_{t-1} - e_t)/e_0 \tag{5}$$
$$e_t := \|u_t - \hat{u}_t\|_2 \tag{6}$$

2. Surrogate reward

$$r_t := \|\hat{u}_{t,\text{refine}} - \hat{u}_{t,\text{no-refine}}\|_2 \tag{7}$$

We used the first reward definition for static and advection experiments where analytic true solutions are available, and for Burgers experiments involving a single initial condition and pre-computed reference data that acts as the true solution. We used the second reward definition for all other Burgers experiments.

## A.2 IMPLEMENTATION

We used standard policy gradient (Sutton et al., 2000) for all experiments except for experiments on Burgers equation and generalization of 8x8-trained advection policies to 64x64 test meshes. We used PPO (Schulman et al., 2017) for the latter two cases. We trained for 20k episodes on static problems, 10k episodes on advection problems, 2k episodes on Burgers equation with a single IC, and 4k episodes on Burgers equation with random ICs. The Burgers experiment with pretraining used 2k episodes on a single IC and a further 2k on random ICs. Each episode is initialized with refinement budget $B$, where $B = 10$ for static problems, $B = 20$ for advection, and $B = 50$ for Burgers.

**IPN.** For efficient computation on a batch of $B$ trajectories, where each trajectory $b$ consists of $T$ environment steps and each step $t_b$ consists of a variable-sized global state $s \in \mathbb{R}^{N_{t_b} \times d}$, we merge the variable dimension with the batch and time dimension to form an input matrix whose dimensions are $[\sum_{b=1}^{B} \sum_{t=1}^{T} N_{t_b}, d]$. The output is reshaped into a "ragged" matrix of logits with dimensions $[B \times T, N_{t_b}]$, where the row lengths vary for each batch and time step. A softmax operation over each row produces the final action probabilities at each step.

**Graphnet policy.** The first graph layer is an Independent recurrent block that passes the input node tensors through a convolutional layer followed by a fully-connected layer, to arrive at node embeddings. This is followed by two recurrent passes through an InterationNetwork (Battaglia et al., 2016) where fully-connected layers are used for edge and node update functions. A final InteractionNetwork output layer followed by a global softmax over the graph produces a scalar at each node, which is interpreted as the probability of selecting the corresponding element for refinement. Except for the input node feature $v^i \in \mathbb{R}^d$ and output node scalar, all internal node (edge) embeddings have the same size, denoted as $\dim(v)$ ($\dim(e)$). We fixed $\dim(e) = 16$ for both static and advection and tuned $\dim(v)$ (Table 2).

**Hypernet policy.** We fixed the main network's hidden layer dimension at $h = 64$ and tuned the hypernetwork's hidden layer dimension $h_1$ (Table 2).

## A.3 HYPERPARAMETERS

For both static and advection problems, we tuned a subset of all hyperparameters for all methods by the following procedure to handle the large set of policy architectures and ground truth functions. Chosen values of tuned hyperparameters are given in Table 2; all other hyperparameters have the same values for all methods and are listed below. We conducted tuning in a multi-task setup, where we train a single policy on functions randomly sampled from all ground truth classes, with randomly sampled parameters according to Appendix A.1.2. This is done separately on static and advection

problems. The tuning process is coordinate descent where the best parameter from one sweep is used for the next sweep. We start with exploration decay $\epsilon_{\text{div}} \in \{100, 500, 1000, 5000\}$ (a lower bound on exploration was enforced by using behavioral policy $\tilde{\pi}(a_t|s_t) = (1 - \epsilon)\pi(a_t|s_t) + \epsilon/N_t$ with $\epsilon$ decaying linearly from $\epsilon_{\text{start}}$ to $\epsilon_{\text{end}}$ by $\epsilon_{\text{div}}$ episodes). Next we tune the size of hidden layers in the policy network (over $(h_1, h_2) \in \{(128, 64), (256, 64), (128, 128), (256, 256)\}$ for IPN, node representation dimension $\dim(v) \in \{32, 64, 128, 256\}$ for Graphnet, and $h_1 \in \{16, 32, 64, 128\}$ for Hypernet). Lastly, we tune the learning rate $\alpha \in \{5 \cdot 10^{-5}, 10^{-4}, 5 \cdot 10^{-4}, 10^{-3}, 5 \cdot 10^{-3}\}$. For Graphnet and Hypernet, we inherit the best $\epsilon_{\text{div}}$ from IPN because optimal exploration depends in large part on the complexity of the environment, which is the same across all policy architectures.

Separately for the static and advection cases, all three policy architectures have the same values for all other hyperparameters. These are: policy gradient batch size 8, initial exploration lower bound $\epsilon_{\text{start}} = 0.5$, final exploration lower bound $\epsilon_{\text{end}} = 0.05$, discount factor $\gamma = 0.99$, convolutional neural network layer with 6 filters of size $(5, 5)$ and stride $(2, 2)$.

Table 2: Hyperparameters for IPN, Graphnet and Hypernet policies on static and advection AMR.

| Parameter | Static | | | Advection | | |
|---|---|---|---|---|---|---|
| | IPN | Graphnet | Hypernet | IPN | Graphnet | Hypernet |
| $\epsilon_{\text{div}}$ | 500 | 500 | 500 | 100 | 100 | 100 |
| IPN $(h_1, h_2)$ | (128, 64) | - | - | (256,256) | - | - |
| Graphnet $\dim(v)$ | - | 64 | - | - | 256 | - |
| Hypernet $h_1$ | - | - | 128 | - | - | 32 |
| $\alpha$ | $10^{-4}$ | $10^{-4}$ | $5 \cdot 10^{-5}$ | $10^{-4}$ | $10^{-4}$ | $10^{-4}$ |

For Burgers experiments and advection experiments on generalization from $8 \times 8$ to $64 \times 64$ initial mesh sizes, we used a more comprehensive population-based hyperparameter search with successive elimination for all methods. We start with a batch of $n_{\text{batch}}$ tuples, where each tuple is a combination of hyperparameter values, with each value sampled either log-uniformly from a continuous range or uniformly from a discrete set. We train independently with each tuple for $n_{\text{episode}}$ episodes, eliminate the lower half of the batch based on their final performance, then initialize the next set of $n_{\text{episode}}$ episodes with the current models for the remaining tuples. We use the hyperparameters of the last surviving model. Chosen values are shown in Table 3.

The hyperparameter ranges are: discount factor in $\{0.1, 0.5, 0.99\}$, policy entropy coefficient in $(10^{-3}, 1.0)$, GAE $\lambda$ in $\{0.85, 0.90, 0.95\}$, learning rate in $(10^{-5}, 5 \cdot 10^{-3})$, PPO $\epsilon$ in $(0.01, 0.5)$, value loss coefficient in $\{0.1, 0.5, 1.0\}$, IPN $h_1$ in $\{128, 256\}$, and IPN $h_2$ in $\{64, 128, 256\}$.

Table 3: Hyperparameters for advection size $\uparrow$ and Burgers experiments

| Parameter | Advection (IPN) | | Burgers (IPN) | |
|---|---|---|---|---|
| | Bumps | Circles | 1 IC | Random IC |
| Discount $\gamma$ | 0.99 | 0.99 | 0.1 | 0.5 |
| Entropy coefficient | 0.0133 | 0.0689 | $8.84 \cdot 10^{-3}$ | $1.17 \cdot 10^{-3}$ |
| GAE $\lambda$ | 0.95 | 0.9 | 0.85 | 0.85 |
| Learning rate | $1.18 \cdot 10^{-3}$ | $4.8 \cdot 10^{-3}$ | $1.59 \cdot 10^{-3}$ | $2.11 \cdot 10^{-4}$ |
| PPO $\epsilon$ | 0.0113 | 0.195 | 0.128 | 0.169 |
| Value loss coefficient | 0.1 | 0.5 | 0.5 | 0.1 |
| IPN $h_1$ | 128 | 256 | 256 | 128 |
| IPN $h_2$ | 128 | 128 | 128 | 256 |

## A.4 COMPUTING INFRASTRUCTURE AND RUNTIME

Experiments were run on Intel 8-core Xeon E5-2670 CPUs, using one core for each independent policy training session. Average training time with 20k episodes in the static case was approximately 6 hours for IPN and Hypernet, and 9 hours for Graphnet. Average training time with 10k episodes

in the advection case was approximately 14 hours for IPN and Hypernet, and 18 hours for Graphnet. Policy decision times are shown in Table 4.

Table 4: Mean (standard error) time in milliseconds per refinement decision on $8 \times 8$ and $16 \times 16$ initial mesh partitions.

|          | $8 \times 8$ | $16 \times 16$ | $24 \times 24$ |
|----------|--------------|----------------|----------------|
| IPN      | 3.22 (0.07)  | 5.85 (0.05)    | 9.64 (0.28)    |
| Graphnet | 7.74 (0.33)  | 13.9 (0.43)    | 23.7 (0.19)    |
| Hypernet | 8.08 (0.08)  | 10.7 (0.05)    | 14.3 (0.17)    |
| ZZ       | 1.96 (0.01)  | 6.94 (0.01)    | 15.5 (0.05)    |

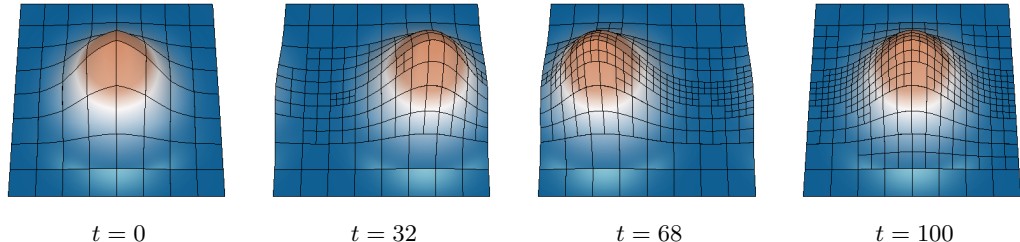

$t = 0$ $\qquad\qquad$ $t = 32$ $\qquad\qquad$ $t = 68$ $\qquad\qquad$ $t = 100$

Figure 7: Advection of a bump function. RL policy trained with budget $B = 20$ generalizes to $B = 100$.

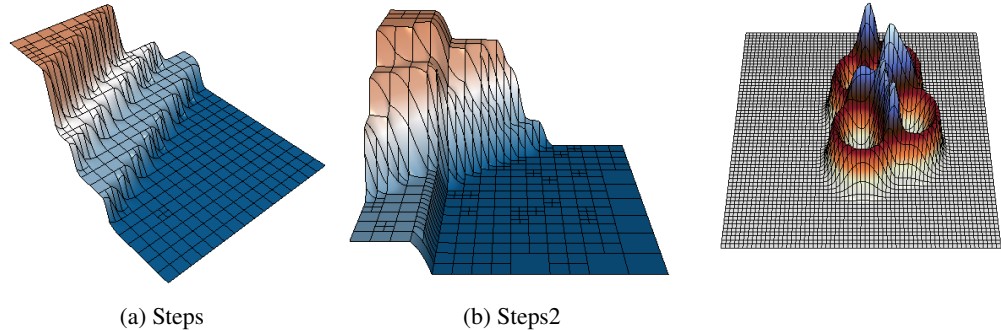

(a) Steps $\qquad\qquad\qquad$ (b) Steps2

Figure 8: Generalization of policies trained with refinement budget $B = 10$ to test case with $B = 100$.

Figure 9: Example test case on $64 \times 64$ mesh

# B  ADDITIONAL RESULTS

**OOD.** In the static case (Figures 13a to 13c), IPN policies trained on *circles* transfer well to *bumps* (and vice versa). Hypernet policies performed poorly overall even in the case of **in-distribution**, and consequently does not show comparable performance when transferring across function classes. In the advection case (Figures 13d to 13f), both IPN and Graphnet policies trained on *bumps* significantly outperformed ZZ when tested on *circles* (compare to ZZ in Figure 3b).

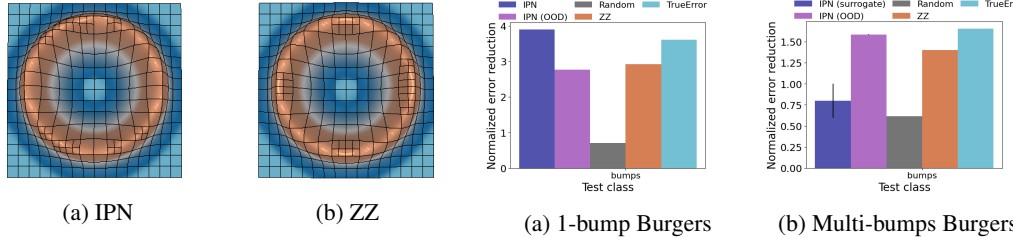

(a) IPN         (b) ZZ         (a) 1-bump Burgers    (b) Multi-bumps Burgers

Figure 10: IPN trained on $8 \times 8$ initial mesh underperformed ZZ when tested on $16 \times 16$ initial mesh but makes qualitatively correct refinements.

Figure 11: **OOD**. (a) IPN (OOD) was trained on Burgers with multi-bumps IC and tested on Burgers with a 1-bump IC. (b) IPN (OOD) was trained on 1-bump IC and tested with multi-bumps IC.

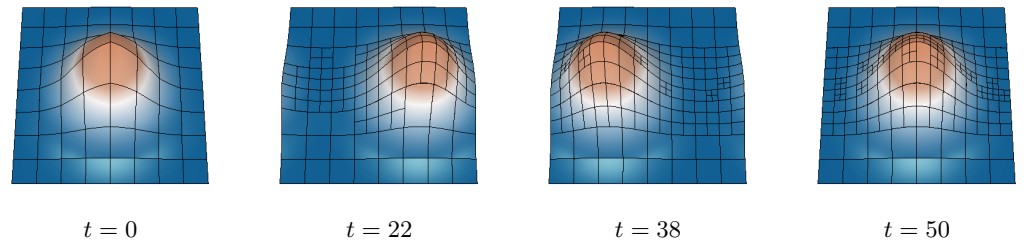

$t = 0$        $t = 22$        $t = 38$        $t = 50$

Figure 12: **Static→advection** and **Budget↑**: IPN trained on static bumps ($B = 10$) transfers to advection ($B = 50$).

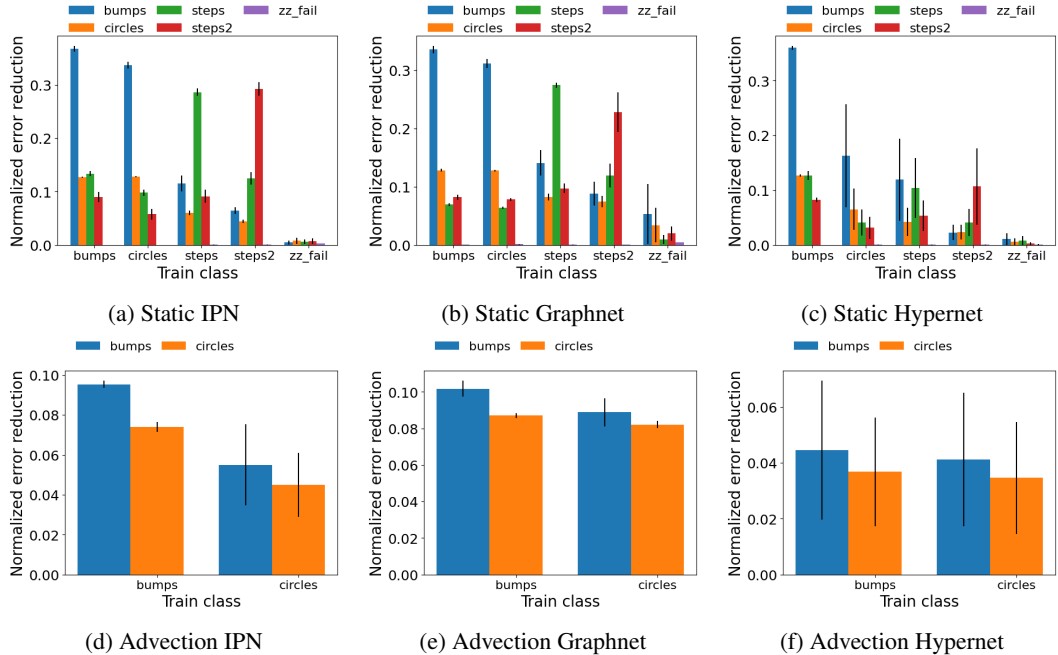

(a) Static IPN      (b) Static Graphnet      (c) Static Hypernet

(d) Advection IPN     (e) Advection Graphnet     (f) Advection Hypernet

Figure 13: All train-test combinations. Normalized error reduction of IPN, Graphnet and Hypernetwork policies on (a-c) Static AMR and (d-f) Advection PDE. Higher values are better. Legend (colors) shows test classes. RL policies were trained and tested on each combination of true solutions. Mean and standard error over four RNG seeds of mean final error over 100 test episodes per method.

