# OpenReview forum: "Reinforcement Learning for Adaptive Mesh Refinement"
_ICLR.cc/2022/Conference — ICLR 2022 Submitted_

### Official Review · Reviewer_LhZp · 2021-11-02

**Correctness:** 4
**Technical Novelty And Significance:** 2
**Empirical Novelty And Significance:** 2
**Recommendation:** 5
**Confidence:** 3

**Main Review:**

Q1. "Nonetheless, to our knowledge, such variation in state-action spaces does not occur in any existing RL application." This comment is not valid. There are many RL formulations and applications that can generalize over a varying dimension of states and actions. Refers RL with neural graph networks and includes these references. For example, many RL approaches seek to solve the solution of various optimization problems formulated as graphs. These approaches train the policy using RL with the small-sized problem and then apply trained policy to solve large-scale unseen problems.

Q2. Is action defined as selecting a computational grid among all the grid points under the current mesh strategy? The proposed action definition sounds inefficient since only one grid point can be selected at each iteration. Is there any reason or justification for using this action definition?

Q3. Why is the simple policy gradient algorithm used? Is there any room for improving the performance by employing more advanced or problem-specific learning algorithms?

Q4. Why independent policy networks (section 4.1) and Hypernetwork policy (section 4.2) are discussed in the main text? Obliviously these policy structures should be inferior to Graph neural network-based policy, and the primary proposed model is GNN based model.

Q5. Which modeling component contributes the most to the generalization capability? For example, state representation and the representation learning, specific reward form, or specific learning strategy?

Q6. How are the computational times compared both in training and test phases?


**Summary Of The Paper:**

The current study formulates adaptive mesh refinement (AMR) as a Markov decision process and applies deep reinforcement learning (RL) to train refinement policies directly from simulation. The proposed method achieved high accuracy while reducing the computational time.

**Summary Of The Review:**

Formulating AMR as an MDP sounds reasonable, and structuring meshing policy using the graph neural network is also straightforward. Formulating a specific component (step) of an iterative algorithm using a learning-based module (i.e., RL) and GNN to learn a policy that can generalize over varying sizes and actions is a pervasive idea. Thus, the technical novelties of the proposed method are limited. The only novelty would be employing these ideas to finite element analysis.

---

> ### Author Response · Authors · 2021-11-14
> **We address Reviewer LhZp's questions on 1) novelty and significance of RL for AMR, 2) variation of state/action spaces in previous work, 3) number of refinements per time step, 4) use of policy gradient, 5) justification for investigating multiple policy architectures, 6) the reason for good generalization performance, 7) train and test times**
>
> We appreciate Reviewer LhZp's time spent on the review and detailed questions, which we address here.
>
> Firstly, we respond to the reviewer’s viewpoint that “the only novelty would be employing these ideas to finite element analysis.” It is a fact that the importance of the entire field of machine learning depends on the importance of the kinds of problems that can be tackled by machine learning methods. Without the ImageNet competition, CNNs would not have become popular; without Atari games, there would be no DQN. It should be clear that finite element methods and the problem of adaptive mesh refinement is at least as important as these established benchmarks, and arguably of greater societal impact due to their prevalence in computational science and engineering. Therefore, the quality of a research paper in ML should not only be measured in terms of theoretical, algorithmic, or architectural advances that are demonstrated on previously-accepted benchmarks. On the contrary, it is crucial to the continued growth of ML to branch out: to demonstrate the strength and generality of learning-based methods on tasks that have traditionally been tackled by decades, if not centuries, of human labor. Yes, the reviewer is correct that the only novelty in our paper is the application of RL to AMR, and nowhere in this paper do we claim to do more than that. However, we sense in the reviewer’s tone that the importance and significance of a first paper on RL for AMR has not been sufficiently appreciated. In our view, this is a significant milestone that opens the door to AMR strategies that are significantly more advanced and powerful than what is practical with experts-driven heuristics. Specifically, an RL approach can enable 1) anticipatory behavior that provides benefit in subsequent simulation steps, which is impossible under a greedy approach; and 2) the integration of $h$-, $p$-, and $r$-refinement into a unified strategy, which does not currently exist via traditional methods that require patching a set of heuristics. While this paper does not give a conclusive solution to these two long-standing grand challenge problems, it serves as a necessary milestone towards solving them in which we developed and demonstrated the feasibility of a foundational RL-based approach that a) is trained using a reward signal rather than human expertise, b) is extensible to real-world FEM problems, and c) performs well on problem instances not seen in training. We hope that the reviewer can reconsider the significance of this paper.
>
> Next, we respond to each specific question in detail:
> 1. We admit that the quoted sentence was not precise enough and removed it. Our intended meaning is better expressed in the last paragraph of Section 2, where we say that RL has been “typically applied to environments with fixed-size observation and small bounded action spaces in almost all benchmark problems”, with an emphasis on benchmark problems such as Atari games, MuJoCo control, and OpenAI Gym environments. Hence, we have removed the offending sentence. We acknowledge the reviewer’s point that applications of RL to graph-structured problems typically have to handle states with increasing size (e.g., if the entire graph is a state). However, in many works in the intersection of graphnets and RL, the action space does not grow with the size of the graph: for example, You et al. 2018 have a fixed set of available atom types from which the policy selects an action, while Trivedi et al. 2020 has a fixed action set consisting of all nodes in the given graph.
> 2. Regarding the remark that “only one grid point can be selected at each iteration” (i.e., MDP time step): yes, as mentioned in our conclusion, this is a limitation of the single-agent RL approach in this paper, which can be overcome by taking a multi-agent perspective whereby each element is an agent who takes refinement actions concurrently with all other agents.
> 3. Regarding using the “simple policy gradient algorithm”, as mentioned in Appendix A.2, we used both standard policy gradient (PG) and proximal policy gradient (PPO) in different subsets of our experiments. Of course, performance may be expected to improve if the base RL algorithm is stronger. The main point is that all three policy architectures are compatible with any policy optimization method in RL, so we believe that using PG and PPO for proof-of-concept is sufficient.

---

> > ### Author Response · Authors · 2021-11-14
> > **(continued) reply to Reviewer LhZp**
> >
> > 4. Regarding “why independent policy networks (section 4.1) and Hypernetwork policy (section 4.2) are discussed in the main text”: we have two answers. Firstly, it is widely-known that different neural network architectures can have significantly different performance even when used in the same learning algorithm (RL, in this case). Secondly, and more importantly, being the first work to apply deep reinforcement learning to AMR, it is incumbent upon us to explore the landscape and show experimental results of diverse design options (the three architectures) on diverse tasks (static, advection, static->advection, budget increase, size increase, Burgers, different boundary conditions). Otherwise, we would not have discovered that the IPN and Graphnet are actually comparable on static problems, that Graphnet is significantly better at generalizing to higher budgets in advection, that Hypernet has high variance and is difficult to train, and that IPN achieves a good tradeoff between ease of implementation and performance.
> > 5. Regarding “which modeling component contributes the most to the generalization capability,” the short answer is that these policy architectures make decisions based on local features, and many problems of interest in AMR all share the same local features. Firstly, note that different problems may look different globally (e.g., different ways of superimposing step functions), but they may have the same local features (e.g., step discontinuities). Our proposed architectures enable the agent to learn a map from local element features to element scores (i.e, logits), from which a global softmax then produces the action probabilities. Hence, the architectures enable a trained policy to generalize well on a test mesh of different sizes and with different budget constraints, because the local element features on the test mesh continue to resemble local element features seen during training (e.g., a discontinuity at a step or a smooth bump), even when the global function on the test mesh was not seen in training.
> > 6. Regarding “computational times compared both in training and test phases”, we reported training times in Appendix A.4 and test times in Table 4, where we see that test runtime of all policy architectures are comparable and have similar magnitude to the ZZ baseline. The end-to-end RL approach of optimizing policies directly with experience in simulation may provide a faster path toward new refinement policies that can integrate $h$-, $p$-, and $r$-refinement into a unified strategy, whereas developing new AMR techniques is very human-labor intensive, so the training time of RL policies would be a negligible expense. Given a high-performing trained policy, the time incurred in training is also negligible when amortized over the all future deployment of that policy.

---

> > > ### Comment · Reviewer_LhZp · 2021-11-21
> > > **Thank you for your responses.**
> > >
> > > Thanks for the detailed reply. I also believe that research on improving performance using ML on critical issues in specific fields is very important. However, rather than simply formulating the problem as MDP and trying to solve it with RL, I think it would be better to have methodological innovation that overcomes the technical difficulties that arise when solving the target problem with ML. For example, suppose the paper suggests a strategy to effectively process and analyze local features or increase the dimension of action. In that case, the paper will be more highly evaluated not only by the domain engineer but also ML researchers.
> > >
> > > Thank for answering to my other questions. I'm going to raise the score in recognition of the author's new attempts and aspects that have produced early results.

---

> > > > ### Author Response · Authors · 2021-11-22
> > > > **Sounds good**
> > > >
> > > > We accept Reviewer LhZp's advice on making strong contributions at the intersection of other fields and ML. We believe that our promising experimental findings---such as outperforming the oracle baseline TrueError, generalizing from a training mesh with 576 nodes to a test mesh with 360000 nodes, and generalizing to unseen initial conditions---are direct outcomes of overcoming the relevant technical difficulties in adaptive mesh refinement. We believe that the issues of processing local features and increasing size of action space are effectively addressed by the candidate architectures that map from a collection of local element features to a global output of element selection probabilities. Having found such promising experimental results, we were under the impression that there was no need at the present time to further complicate the technical approach. We can understand that Reviewer LhZp's acceptance criteria requires more algorithmic and architectural novelty. We will keep this in mind for future submissions to ML venues.

---

### Official Review · Reviewer_aatv · 2021-11-05

**Correctness:** 3
**Technical Novelty And Significance:** 2
**Empirical Novelty And Significance:** 3
**Recommendation:** 5
**Confidence:** 3

**Main Review:**

This paper is mostly clearly written, and has a conceptually simple yet clear idea behind it. At its core, it proposes to use reinforcement learning to train neural networks to propose adaptive mesh refinement operations in order to reduce simulation error (in contrast to traditional methods, which tend to use heuristics to perform the adaptive refinement). Therefore, this work's evaluation rests mainly in the empirical demonstration of the benefits of the proposed method.

However, the practical implementation of this simple becomes more convoluted, and is one of the weak points of the paper. Three variants for the neural network architecture are proposed, a simple "per-node" architecture, a hypernetwork based one and a graph-net based one. It is not necessarily an issue to propose diverse variants of a single method. However, in this case there is not much justification as to why different methods are tried, or how one would choose one over the other when applying the proposed approach in practice.
In the end, the results are presented in such a way that the best performing of the 3 proposed variants is presented as better than baselines, though no single one of the 3 is able to consistently perform best. Given that there is no analysis or proposed method to pick a priori which one of the variants is more appropriate for a given task, the results become muddied.

Nevertheless, the so called "Independent Policy Network" (IPN) seems to be the generally best performing variant, usually performing better or on par with the baselines, though not always. It is important to take into account that training the neural network model incurs some considerable cost, and as such performing on par with a traditional method might not be enough.

To this end it would also be important to have comparisons of run-times between the methods and baselines, not only error reductions. This is required in order to evaluate if the proposed mesh refinements are actually reducing error by being "smart" about the refinement, instead of simply proposing "more" refinement, or more intense computation.

Some additional questions and comments:

- Even though the authors state it should be simple to implement, the policy networks in this work do not have a "coarsen" or "de-refine" action. Does this mean the meshes simply get finer and finer throughout the simulation? How does this impact long simulations, and ones where for example a shock wave is moving (and thus the region that requires refinement is moving)? Would this lead to an overall very fine mesh? This is important in light of the comment above, regarding comparisons of run time of the proposed method and baselines

- For the IPN, the description is a bit confusing. At one point it is stated that it does not depend on other elements, yet at a later point it is said that information of surrounding elements is included. It was also unclear to me how the convolutions are applied here. It would be good to improve this description.

----------

After reading the authors response and discussing the issues above with the authors (see discussion in the thread below), I am still not fully convinced of the robustness and strength of the experimental evaluation. As such, I will maintain my previous evaluation.

**Summary Of The Paper:**

This paper proposes using reinforcement learning algorithms to train a few variants of neural network-based models to perform adaptive mesh refinement. These models predict for each node in a mesh where a refinement operation should be performed.

**Summary Of The Review:**

Overall, given the concerns above, I do not feel like this work is ready for acceptance at this point. This is a work that rests mostly on its empirical results, and yet these are not completely robust at this point, given the issues described previously. As such, I am classifying as marginally below acceptance for now. Nevertheless, I am interested in reading the authors responses to the points raised above.

---

> ### Author Response · Authors · 2021-11-14
> **We address Reviewer aatv's questions on 1) justification for investigating different policy architectures, 2) whether the learned refinement policies are taking "smart" actions, 3) justification of the training time for an RL approach, 4) derefinement, 5) description of IPN and use of convolutional networks**
>
> We appreciate Reviewer aatv's time spent on the review and detailed questions, which we address here:
> 1. Regarding “justification as to why different methods are tried”, each of the subsections 4.1, 4.2, 4.3 contain a dedicated paragraph that explains the limitation and advantage of each of the three candidate policy architectures. To quote from the text, the IPN “makes a strong assumption of locality as the action probability at an element does not depend on the observations at other elements,” while its property of permutation equivariance means that “one cannot use the ordering of inputs to represent spatial relations among elements.” These limitations directly motivate the construction of the Hypernetwork policy, which “captures higher-order interaction among inputs” by defining the main network weights as a function of the inputs. However, this means the hypernetwork lacks “an inductive bias for the local nature of interactions seen in classical applications of AMR,” while the fact that “complete global information affects each local refinement decision” is an extremely strong inductive bias. These limitations are resolved by the Graph Network policy, which contains the advantage that “cross terms arise in the forward pass”, hence retaining the hypernetwork’s advantage, and that “local spatial relations between mesh elements are included by construction in the initial edge attributes”, which addresses the hypernetwork’s disadvantage.
>     - 1a. One may wonder why this paper even discusses the IPN and Hypernetwork architectures when the Graphnet seems perfectly suited to the AMR task. We have two answers. Firstly, it is widely-known that different neural network architectures can have significantly different performance even when used in the same learning algorithm (RL, in this case). Secondly, and more importantly, being the first work to apply deep reinforcement learning to AMR, it is incumbent upon us to explore the landscape and show experimental results of diverse design options (the three architectures) on diverse tasks (static, advection, static->advection, budget increase, size increase, Burgers, different initial conditions). Otherwise, we would not have discovered that the IPN and Graphnet are actually comparable on static problems, that Graphnet is significantly better at generalizing to higher budgets in advection, that Hypernet has high variance and is difficult to train, and that IPN achieves a good tradeoff between ease of implementation and performance. We believe that our experimental results provide a solid building block for follow-up work to answer the question of how “one would choose one over the other when applying the proposed approach in practice.”
> 2. Regarding whether “the proposed mesh refinements are actually reducing error by being ‘smart’ about the refinement, instead of simply proposing ‘more’ refinement, or more intense computation,” we emphasize that all methods were subject to the same refinement budget constraint, meaning that no method was allowed to take more refinement actions than other could. As for “smartness” of refinement, we draw attention to two facts: 1) our method performs significantly better than the Random baseline in every single experiment; 2) our submitted paper contains a link to comprehensive videos of the refinement policies in action at this link https://sites.google.com/view/iclr2022-amr, and we hope that the reviewer can see from these videos that the learned policies are “smart”, e.g. anticipating and tracking a moving feature that requires refinement.
> 3. Regarding “comparisons of run-times between the methods and baselines,” we show in Table 4 (Appendix, page 17), that all test runtimes are comparable and within the same order of magnitude. The training time is also reported in Appendix A.4. The one-shot training time for RL refinement policies can be justified by the fact that: 1) Given a high-performing trained policy, the time incurred in training is also negligible when amortized over the all future deployment of that policy; 2) The end-to-end RL approach of optimizing policies directly with experience in simulation may provide a faster path toward new refinement policies that can integrate $h$-, $p$-, and $r$-refinement into a unified strategy, whereas developing new AMR techniques is very human-labor intensive, so the training time of RL policies would be a negligible expense.

---

> > ### Author Response · Authors · 2021-11-14
> > **(continued) Reply to Reviewer aatv**
> >
> > 4. Regarding whether “the meshes simply get finer and finer throughout the simulation.” Yes, that is correct. It is not a limitation of the method, but a limitation of the scope of this particular paper. As we have stated in the paper, all three proposed architectures can be easily extended to include derefinement actions. Our videos on Burgers equation at this link https://sites.google.com/view/iclr2022-amr already shows that the refinement “front” anticipates and tracks the movement of features. It is reasonable to expect that a policy that includes derefinement and refinement can track a moving region that requires refinement.
> > 5. The reviewer’s question about the IPN is actually a question about the definition of the observation $o^i$ associated with each element $i$. This is described in detail in Appendix A.1.1. To recapitulate, each element observation $o^i$ is a $l \times w \times c$ image, which represents a spatial window that, in general, can consist of gridpoints sampled both within and outside the element $i$. If there are sampled points outside the element, this means that the observations $o^i$ and $o^j$ of adjacent elements $i$ and $j$ have a certain amount of overlap. If there is such an overlap, then the IPN may capture some dependency between adjacent elements. However, if there is no overlap (one chooses not to sample points outside the element), then the IPN truly treats each element independently from all others. Regarding convolutions, the fact that each $o^i$ is an image tensor means that the first layer of the IPN network is a convolutional network that applies the CNN filter over the image.

---

> > > ### Comment · Reviewer_aatv · 2021-11-22
> > > **Response**
> > >
> > > 1. Even though there are "dedicated paragraph that explains the limitation and advantage of each of the three candidate policy architectures", it is not possible to take from these paragraphs precisely when to apply each in practice. Given a new problem, which architecture should I use? How to know a priori which will perform better? Here the authors seem to claim that the Graph Network model is the "strongest", or at least the final in the chain of development of the proposed methods. Yet in many experiments the IPN outperforms the Graph Network based model, and in a couple experiments the results for the Graph Network are not presented (just IPN).
> > >
> > > 2. I understand that all methods have the same refinement budget. Yet does that mean all methods end up with the exact same number of mesh nodes for all time points of the simulation? Is that what is being claimed? From my understanding a budget could simply not be fully used, or be used at different time points, leading one method or another to use more nodes per timepoint at the end of a simulation. Correct me if this understanding is wrong and indeed all methods are using exactly the same number of nodes per timestep.
> > >
> > > 3. Related to the previous point (2) as well, the table in the Appendix only gives information on run time "per refinement decision". Yet it would be important to have *total* runtime for the tasks at hand, given that number of decisions are possibly different, and that which particular decisions are taken can influence solver time. There is no reason not to include full runtime, as this would be a valuable and insightful information on the performance of the proposed method versus the baselines. Moreover, even the information that is included in the appendix is mostly not favorable to the proposed method. The Graph Net, which as mentioned previously is supposed to be the strongest method, performs significantly worse than baselines for per decision runtime in all cases. Additionally, the other methods also take longer in some of the cases.
> > >
> > > 4. When comparing to the baselines, are the baselines allowed to "de-refine"? If not this is a severe limitation of the baselines. If they are, then this implies that, as mentioned in item (1), the baselines might end up using a less fine mesh on average (implying lesser total run time). Even though it might be "conceptually simple" to add a de-refinement action to the proposed method, given no empirical evidence to support this idea's functionality, we cannot assume positive results as of now.
> > >
> > > 5. It would be useful to clarify these ideas in the text, as previously mentioned.
> > >
> > > Given the issues above, I maintain my previously raised concerns. If the authors would like to further address these, I am open to reading their reply.

---

> > > > ### Author Response · Authors · 2021-11-23
> > > > **We address Reviewer aatv's questions on 1) which architecture to use for test problems, 2) whether different methods have the same budget and number of refinements, 3) total runtime, 4) de-refinement**
> > > >
> > > > We appreciate Reviewer aatv's time spent on reading our rebuttal and asking detailed follow-up questions, which we address here:
> > > >
> > > > 1. Our first round of rebuttal addressed Reviewer aatv’s question about “justification as to why different methods are tried”. Our answer to the question of “Given a new problem, which architecture should I use?” is the following: choose the architecture that has the highest performance on the type of local features that is likely to arise in the test problem. For example, our results on static problems in Figures 3a, 5a, and 6a strongly and clearly show that the IPN is the best candidate for test problems whose local features are stationary or slowly-propagating. Moreover, our results on the advection problem in Figures 3b, 5b, and 6b clearly show that the Graphnet policy is the best candidate for test problems whose local features are smooth variations that propagate at a non-trivial velocity. One can anticipate the type of local features that will arise in test problems based on domain experience. Many of our anticipated use cases for an RL approach have a long history of being solved by traditional methods, and hence there is a wealth of prior knowledge on the kinds of features that will arise.
> > > > 2. Yes, all methods have the same refinement budget. Yes, having combed through our extensive set of training and test results, we can confidently say that, with very few exceptions, all methods end up with the exact same number of mesh nodes for all time points of the simulation. The only exceptions we found are those for the Hypernet policy in the case of static-circles and advection-circles, and IPN on advection circles, where certain training runs failed to improve reward and ended up making no refinement at all. This explains the poor performance of Hypernet on static- and advection- circles and IPN on advection circles. All other cases fully used the entire budget, meaning that one refinement was taken at each time step of an episode.
> > > > 3. Relating to the preceding response in #2, the total number of refinement actions are almost always the same. Therefore, one can recover total runtime used in a test episode by multiplying the average time per refinement by the number of steps in a test episode. Regarding the possibility that “which particular decisions are taken can influence solver time,” we point out that the timing information in Table 4 is the mean over (10 episodes) * (10 steps per episode), which accounts for the impact of variability of different element refinements on the solver time. We point out that the mean runtime carries the same information content as the full runtime, since the number of steps is known. Appendix A.4 reports the total training time.
> > > > 4. To ensure a fair comparison in the context of refinement only, neither our method nor any baseline is allowed to de-refine. The lack of refinement is not a limitation of our method, and it is also not a limitation of any baseline. The extension of our method to de-refinement is to 1) map each input observation $o^i$ to two output nodes, one interpreted as refinement and the other de-refinement; 2) apply the same global softmax to the collection of output nodes; 3) select an element for refinement or de-refinement action based on which node and which type of node was sampled.
> > > > 5. We will include more details about the explanation of the observation in the main text in our next revision.

---

> > > > > ### Comment · Reviewer_aatv · 2021-11-23
> > > > > **Response**
> > > > >
> > > > > A brief comment on 3: as I understand it, it is not correct that "the mean runtime carries the same information content as the full runtime", since what is provided is the mean time per refinement operation, which is different than the full runtime of solving the equation (which includes not only refining, but also the time for the solver). This is the quantity I would have liked to see reported, as it is what matters in the end.
> > > > >
> > > > > For 4, is it fair to have baselines that are not able to de-refine? In practice the methods that are used for AMR would be able to de-refine, and that would allow for more efficient execution, affecting, for example, the total run times I mentioned above.
> > > > >
> > > > > As for 2, does this imply that the baselines are also limited to performing only 1 refinement operation per step? Since you have stated that the baselines end up with the same number of nodes as the proposed method in every step.

---

> > > > > > ### Author Response · Authors · 2021-11-24
> > > > > > **Additional Response**
> > > > > >
> > > > > > For all of these points, we would like to clarify that the RL policy and the baselines are all evaluated under the same conditions and constraints: the same initial mesh and initial conditions are used, one refinement action is performed per step, and the same number of steps is performed. As a result, the meshes for the various approaches have an identical number of elements/nodes at each step, and the only difference between them is the choice of which element to refine.
> > > > > >
> > > > > > With regards to the runtime question, because of the points mentioned above, the differences in runtime between the RL approach and the various baselines manifest only through the differences in the runtime of evaluating the refinement indicator -- the RL policy or the baselines -- since the time required for the PDE solve in between refinement actions is nearly identical between all approaches (as the meshes for all methods are of identical size at each step and there are negligible differences in the computational cost required to solve the PDE on two meshes that differ by only the choice of which elements are refined). Therefore, a comparison between the mean runtime per refinement action versus the full runtime effectively differs by only a constant offset (solver time between refinement actions) and a constant factor (number of refinements per episode).
> > > > > >
> > > > > > With regards to the derefinement question, we believe it to be fair if the baselines and the RL approach are operating under the same constraints: no approaches are allowed to derefine and each approach refines at most one element at each step. We chose these constraints as it is straightforward to evaluate performance by evaluating the error reduction. If other approaches such as derefinement (or a wider variety of options such as p/r refinement, multiple actions per step, or relative budget constraints) are implemented, the evaluation metrics are not as clear cut anymore and require some ad hoc parameterization of relative metrics such as accuracy in comparison to the computational cost of mesh refinement and solver time. We believe that to be out of the scope of a preliminary paper on AMR for RL given the page limit and the amount of experimental results shown, but implementing these additional capabilities is in the development path of this project.

---

> > > > > > > ### Comment · Reviewer_aatv · 2021-11-29
> > > > > > > **Response**
> > > > > > >
> > > > > > > I understand all of the factors you mentioned as to why the total runtime should be directly related to the runtime per refinement action. Even so, I would like to see the direct numbers. I see no reason why to report a proxy metric and make arguments for why it is equivalent, instead of simply providing the total metric (or preferably both, of course). As you say, this is favorable to the proposed method, so it should boost your case to present those numbers directly.
> > > > > > >
> > > > > > > I am not sure it is fair to present only "constrained baselines". Yes it is fair in the sense of a "fair competition" between methods, and a valid comparison. But I also believe it would be important to include baseline methods that are state of the art (or close), so that we can get an idea of how the method stacks against current best methods (which would likely include the option of derefining).
> > > > > > >
> > > > > > > Both of these are still significant issues that I see in the paper.

---

> > > > > > > > ### Author Response · Authors · 2021-11-29
> > > > > > > > **Reply to runtime and de-refinement**
> > > > > > > >
> > > > > > > > As Reviewer aatv has agreed, reporting mean runtime and total runtime are equivalent because they differ only by a constant offset and a constant factor. This implies that the choice to report one or the other is not a significant issue with the paper.
> > > > > > > >
> > > > > > > > We reiterate our belief that a first paper on RL for adaptive mesh refinement, submitted to a machine learning venue, does not benefit from the complications that will arise due to de-refinement. The current paper already has two definitions of reward and two different evaluation metrics just for the case of refinement alone. De-refinement will involve additional evaluation metrics that consist of multiple objectives that are unfamiliar to an ML audience. Such fine-grained domain-specific details will detract from the core message of this paper: RL is good enough at generalization on a diverse set of test problems in comparison to an oracle baseline using the true error, that it is worthwhile for researchers at the intersection of machine learning and computational science and engineering to pursue this research direction.

---

### Official Review · Reviewer_rRgy · 2021-11-06

**Correctness:** 3
**Technical Novelty And Significance:** 4
**Empirical Novelty And Significance:** Not applicable
**Recommendation:** 5
**Confidence:** 3

**Main Review:**

The problem of mesh refinement is critical in engineering, and current practices, as the authors stated, are often designed from some heuristics. The application of RL to this mesh refinement, as far as I know, is novel, promising, and may provide new ideas to solving this open problem. Nevertheless, the authors tested their framework on some toy examples with low resolution and barely sufficient validations.

The PDEs to be solved in practice, as the authors indicated, usually have millions or billions of degrees of freedom. The authors claimed to "show that an RL refinement policy can generalize to higher refinement budgets and larger meshes." However, the "larger meshes" only contain 64x64 (4096) nodes, which is only a tiny fraction of the PDEs to be solved in practice. Hence, it is questionable whether the method may apply to PDEs on a practical scale, showing that the policy is "efficient". In other words, the paper would be more convincing if the authors demonstrate a case where their method is applied to a mesh containing millions of degrees of freedom and still outperform the baselines.

Another concern is about validation. The authors compared their policy with some trivial policy but did not compare with the state-of-the-art in adaptive mesh refinement. Hence it is not convincing that the RL-based method may outperform the traditional "heuristic" methods on this problem. The paper would be more grounded if the authors compare the RL-based method with the latest, non-RL-based method on the same problem set.

**Summary Of The Paper:**

This paper presented an RL-based method to perform adaptive mesh refinement to solve PDEs more accurately.

**Summary Of The Review:**

The paper proposed a promising direction for future research, but it would be more convincing if the authors tested their method on practical scales with more solid validations.

---

> ### Author Response · Authors · 2021-11-14
> **We address Reviewer rRgy's questions on 1) testing on larger meshes, 2) strength of the baselines, 3) size and scope of experimental setup**
>
> We appreciate Reviewer rRgy's time spent on the review and request for more experimental results, which we address here:
>
> 1. We have updated the paper with new experimental results in Figure 6d on page 9 that shows the test of an 8x8-trained IPN policy on a 200x200 test mesh. We see that the trained policy still outperforms the ZZ and TrueError baselines on circle initial conditions, despite the 625x increase in the number of elements. Average times per action by the baselines and the RL policy were on the same order of magnitude. We wish to clarify that 200x200 refers to the number of elements, whereas there are actually 200x200x3x3=360000 nodes. Our FEM setup runs in serial rather than parallel, which makes it difficult to test on much larger meshes, but we hope this result on 360000 nodes addresses the reviewer’s request.
> 2. More generally, in tests of generalization to larger meshes, the relative increase in mesh size is more important than the absolute sizes of the individual meshes. In Figure 6, we showed that a policy trained on merely 8x8 (64) elements generalizes to 64x64 (4096) elements, even outperforming the TrueError baseline in one case (note that TrueError is not applicable to systems with unknown solutions). This is a 64x increase in the number of initial elements. Regardless of whether a 64x64 mesh is considered “small”, this generalization result is significant because it validates a key property of the proposed policy architectures: they can take a good global action at test time on a mesh whose size is vastly different from that training, precisely because the global action is the result of local processing of the (variable-sized) inputs. Our result shows that the policy performs well as long as local features on a test mesh resemble those seen in training. This property holds independently of the size of the test mesh.
> 3. The appearance of simplicity of the baselines may have led the reviewer to characterize them as “trivial”. They may be simple and easy to describe, but they are very strong baselines. We emphasize that the TrueError and GreedyOptimal baselines are oracles. TrueError has access to the true solution, which means it is effectively an upper bound on the performance of any state-of-the-art method that is based on instantaneous error estimation (note: refining the element with highest true error may not lead to the largest error reduction, so it is possible but rare that another error-estimation-based method outperforms TrueError simply due to random chance). GreedyOptimal performs a one-step lookahead by checking all possible outcomes of refining each element individually and chooses the element whose refinement would result in the lowest error at the next step. This is an extremely strong baseline that is intractable in practice for even simpler time-dependent PDEs on relatively coarse meshes.
> 4. We respond to the expectation that the proposed methods should be “applied to a mesh containing millions of degrees of freedom.” We agree that this will serve as direct evidence of the real-world applicability of the RL approach. While we already have plans to scale up to more expensive evaluation, we point out that the scale and scope of our current experimental setup is consistent with previous work at the intersection of FEM/PDE and machine learning that have been accepted at conferences such as ICLR/ICML:
>     - [1] Hsieh et al. ICLR 2019 who experimented with the 2D Poisson equation with a 64x64 square train mesh;
>     - [2] Luz et al. ICML 2020 who experimented on the diffusion PDE on 2D triangular mesh with number of nodes ranging from 1024 to 400k;
>     - [3] Pfaff et al. ICLR 2021 who learned models of simulations with 1k-5k nodes;
>     - [4] Alet et al. ICML 2019 who experimented on the Poisson equation on 7x7 mesh with square and sphere domains;
>     - [5] Belbute-Peres et al. ICML 2020 who experimented on airfioil with 6648 nodes on a fine mesh and 354 nodes on a coarse mesh.

---

### Official Review · Reviewer_Gif3 · 2021-11-07

**Correctness:** 1
**Technical Novelty And Significance:** 3
**Empirical Novelty And Significance:** 2
**Recommendation:** 5
**Confidence:** 4

**Main Review:**

Article review and position the paper in the large context of AMR:
(major) Authors write in the abstract: "Existing scalable AMR methods make heuristic refinement decisions based on instantaneous error estimation and thus do not aim for long-term optimality over an entire simulation." This is a bit strong statement. For example, people in computational science for all finite-element, finite-volume, and spectral finite element (SEM) use mesh sensitivity method for AMR. In such cases, adjoint solver is used to measure the sensitivity of the solution. Although this approach employs some coefficients that are hard to quantify a-priori, however, there is a very solid and concrete argument on how to choose them, as well as their interpretability. Since, tuning the parameters is also part of RL and all three architectures used, I think this part should be removed, which brings me to another discussion. What is the fundamental reason of using RL for AMR compared to established AMR methods in finite-element or SEM community? There exists several differentiable FE, FV, or SEM solvers that would readily give the user the adjoint variables using AD, which in turn, can be used for AMR. The algorithm is automated, and based on calculus of variation so mathematically sound. Also the results are either as good as other AMR methods or even exceeding them (please see Adjoint error estimators and adaptive mesh refinement in Nek5000, among many others). When using adjoint-based AMR, it is also possible to define the cost function for mesh refinement based on the terminal solution instead of instantaneous refinement. So that aspect of the paper is also not necessarily novel in a sense that other class methods can also accomplish that. Furthermore, when it comes to optimal control, the mesh refinement can be done with a specific task of achieving the cost function related to the performance (see Rannacher: model reduction by adaptive discretization in optimal control or Apel et al Graded meshes in optimal control for elliptic partial differential equations: an overview, among others). Such approaches are also very scalable, which based solely on the results of this paper, it's hard to claim. Overall, I think i) some claims are not true, ii) some traditional AMR methods are absent in the analysis and introduction of this paper.

Discussion on test cases and practical aspect of the algorithm:
(major) The test cases considered in this manuscript are toy models that can be served as "proof of concept". However, I'm not convinced that such models can be really considered a proof-of concept. The geometry is simple 2D periodic domain and in almost all practical PDE analysis, this is not the case. One beautiful aspect of the work is Out-of-distribution testing, which for the Burger PDE is done based on IC. This is an important test but what about the cases that BCs are changed? The Burger equation is inviscid, what about viscous Burger, for which the Reynolds number (or viscosity) can be changed. A parametric study of PDEs is always challenging and if authors can show the algorithm is robust to such changes, that would a strong argument for the RL-based AMR. Also, the geometry is so simple and for mesh generation this is really a challenge. One of the main concerns of numerical scientists is the complex domain/field analysis. For example, when there is a boundary layer in vicinity of airfoils. The method should demonstrate itself when it comes to more complex geometries. Otherwise, it hard to justify why use RL instead of traditional methods. Finally, MDP transition P consists of one step in PDE solution. Considering RL may require a large number of trajectories, for complicated PDEs, I imagine the training for all relevant parameters (IC, BCs, parameters such as viscosity, etc.) may be intractable. How do authors respond to that?
(minor) One nice trick in the paper is the use of so-called surrogate reward for training. However, it's not clear to me why such reward is superior to exact reward in results of Figure 4. Can authors clarify that? Why not such comparison for other cases?
(minor) Seems like the GreedyOptimal approach is superior in Fig 3 results. Why not comparing with this AMR technique for the rest of figures?

Discussion on the solution space and mathematical definitions:
(major) The finite element requires a bit of discussion on the mathematical formalism. For example, authors discuss "test function", which has a established meaning in finite element community, but seems like they don't refer to the same concept. The finite element method is a discretization of the weak formulation, for which one use the inner product with test function. The next step is the transformation of such weak formulation from the functional space with infinite basis into the functional space with finite basis Such test functions, all of which belong to appropriate Hilbert spaces that should be mentioned (for example, it could be H1 space). I understand that authors are relating to the ML community, for whom the "test functions" may have a separate meaning but this causes confusion.
(minor) Furthermore, authors use "finite element space", which again is not formally defined.
(major) On a similar note, how do authors enforce continuity within elements, if they belong to H1 space? If this is not necessary, then authors should clarify.

Discussion on the algorithm:
(minor) From Fig 5-d, and the manuscript, I'd say authors use non-conformal elements, which is fine; however, how to enforce conforming mesh? Is it possible to enforce not having hanging nodes? Some finite element solvers can't handle the non-conforming mesh and should be modified.
(major) How about mesh quality check? This is specially true if triangular elements are going to be used, which is likely for complex geometries. How to avoid de-generative elements, or irregular mesh skewness? Seems like a constrained RL framework should be used but that would require changing the three architectures proposed.
(major) Authors mentioned de-refinement (or coarsening) is a future direction of this research. However, almost all traditional AMR algorithms use both refinement (in regions with high sensitivity) and de-refinement (in regions with low sensitivity) to achieve not only an accurate solution but also low computational cost. Dropping the de-refinement in AMR algorithm makes it much less competitive compared to state-of-the-art.




**Summary Of The Paper:**

For various complicated problems governed by PDEs (e.g. solid/fluid interactions, aerodynamics, elasticity, backscattering, etc.) the computational cost can become prohibitive even for one inquiry, let alone parametric study. In the same time, mesh refinement is crucial to achieve acceptable accuracy. To mitigate such challenges, one solution is to use adaptive mesh refinement (AMR), for which the mesh refined only in the regions that numerically are sensitive to error propagation. For example for boundary layer models or shock-boundary layer interactions, one much capture the dynamics in high-gradients region of the solution, which are typically in vicinity of the walls or are part of the solution, while in very far regions of the domain, a coarse mesh is sufficient.
Authors recognize that the process of AMR, refinement at each step, can be formulated a Markov decision process (MDP) and hence utilize reinforcement learning (RL) to train refinement policies directly from simulation. But this in turn poses a new challenge, at each step, the dimension of state (number of elements) and action space may (and should) alter. They propose suitable policy updates to overcome this challenge and come up with three different architectures for the implementation. Three test cases are used for experiments (static, advection and Burger) and authors compare all three architectures with each other as well as some traditional AMR methods to demonstrate the performance of the proposed method. They also carry out extra tests on the same set of PDEs to show generalization and out of distribution capabilities of the method for both static and transient PDEs. The paper is well-written and two tricks for RL are impressive (using Nmax to take care of varying dimension of state/action space and use of surrogate reward for training). However, I have some major reservations that I'll explain below in the Main Review.


**Summary Of The Review:**

The paper is well-written and two tricks for RL are impressive (using Nmax to take care of varying dimension of state/action space and use of surrogate reward for training). However, I have some major reservations: 1) Article review and position the paper in the large context of AMR (the adjoint-based AMR and goal-oriented mesh refinement in the PDE-optimal control community can achieve all goals proposed in the paper), 2) Discussion on test cases and practical aspect of the algorithm (other BCs, parameter should be explored, geometries are too simple for mesh generation whose main backlog is dealing with complex problems, scalability for complicated PDEs like Navier Stokes over complex geometries, etc.), 3) Discussion on the solution space and mathematical definitions (this is rather lighter comment but I think authors should clarify some concepts better in the manuscript), 4) Discussion on the algorithm (what about other constraints of AMR that traditional algorithms can handle, de-refinement, etc.).
Overall, when it comes to RL for AMR I think it is novel and authors use some nice tricks to handle the challenges. But the overall use of RL for AMR over traditional methods is not demonstrated and there remain major concerns regarding the pragmatism of the method.

---

> ### Author Response · Authors · 2021-11-14
> **(continued) reply to Reviewer Gif3**
>
> In response to questions about the surrogate reward, we posit that this is not a general behavior and the better performance of the surrogate reward for this case may be linked to different exploration paths encountered during training. Because the surrogate reward is an upper bound on the true reward and provides a positive reward whenever a refinement action causes a change in the solution, it effectively acts as an “exploration bonus”, which has been observed in the RL literature to improve performance [1]. For brevity in the results, we focus the evaluation of the surrogate reward on the problem where an analytic solution is not readily available. In response to the question about the use of the GreedyOptimal approach, we reiterate that GreedyOptimal is a brute-force lookahead that is completely intractable to compute for even small-scale time-dependent PDEs. Only for the case of a relatively small budget/mesh with an analytic solution (advection) is it feasible to compute this method.
>
> In response to the point about mathematical notation regarding FEM, we will update the manuscript to clarify the definition of test function. In view of the page limit, an in-depth description of the discretization (e.g., variational formulations, Hilbert spaces, etc.) is not largely important to the proposed approach and results. As including these details would require omitting some results, we believe that it is sufficient to refer the reader to works that address these details in-depth. In short, the finite element space is an H1 space for the static problems and is an L2 space for time-dependent problems, both which are represented with second-order Bernstein basis functions as mentioned in Section 5.1. For implementation details regarding continuity, we refer the reviewer to the referenced work (MFEM, Anderson et al., 2021).
>
> In response to the questions about mesh characteristics, these issues (enforcing conforming meshes, avoiding degenerative and irregular elements) are primarily with regards to the choice of FEM solver -- the RL policy is only used for selecting elements. If, for example, one would like to include element skewness as a metric to the RL policy, it is straightforward to add this as a channel in the observation, much like how the element depth is observed in the proposed approach. As mentioned in the manuscript, de-refinement is straightforward to implement in the RL framework, but as a proof of concept of the approach, we believed it to be sufficient to just focus on refinement as it is much more straightforward to compare and evaluate against other baselines.
>
> [1] Tang, Haoran, et al. "# Exploration: a study of count-based exploration for deep reinforcement learning." Proceedings of the 31st International Conference on Neural Information Processing Systems. 2017.

---

> ### Author Response · Authors · 2021-11-14
> **We address Reviewer Gif3's comments on 1) the fundamental reason of using RL for AMR compared to established methods, 2) applicability of adjoint-based methods, 3) the validity of our choice of problems as proof-of-concept, 4) the behavior of the surrogate reward and GreedyOptimal baseline, 5) implementation details regarding FEM, 6) mesh characteristics**
>
> We appreciate Reviewer Gif3's time spent on the review and the detailed questions, which we address here:
>
> We reply to the question about “the fundamental reason of using RL for AMR compared to established AMR methods in finite-element or SEM community”. Our contribution is a proof of feasibility for an entirely novel way of learning AMR strategies using deep RL rather than relying on human experts to develop heuristics, opening the door to eventual AMR strategies that are significantly more advanced and powerful than what is practical with experts-driven heuristics. Specifically, an RL approach can enable 1) anticipatory behavior that provides benefit in subsequent simulation steps, which is impossible under a greedy approach; and 2) the integration of $h$-, $p$-, and $r$-refinement into a unified strategy, which does not currently exist via traditional methods that require patching a set of heuristics. While this paper in its limited space does not give a conclusive solution to these two long-standing grand challenge problems, it serves as a necessary milestone towards solving them in which we developed and demonstrated the feasibility of a foundational RL-based approach that a) is trained using a reward signal rather than human expertise, b) is extensible to real-world FEM problems, and c) performs well on problem instances not seen in training.
>
> In response to the point of adjoint-based error estimators, those approaches are generally only practical for steady or quasi-steady problems where the proposed benefits of RL (anticipatory refinement, long-term optimality, etc.) aren’t applicable -- the problems in the works mentioned by the reviewer would not gain any tangible benefits from performing these types of refinement actions. For the class of problems where these benefits can manifest such as nonlinear hyperbolic systems, the use of an adjoint-based error estimator optimized for “terminal state” error would require a great deal of complexity such as the need for a forward/backward solve, a checkpointing system for the backwards-in-time solution, etc. As such, those approaches are not in fact scalable, particularly not in the context of more complex time-dependent systems, and recent works such as that of Davis and Leveque, 2020 (Analysis and Performance Evaluation of Adjoint-guided Adaptive Mesh Refinement for Linear Hyperbolic PDEs Using Clawpack) utilize relatively coarse grids to compute these problems. In contrast, we attempt to show that RL-based approaches that are trained on small computationally feasible problems can perform well when applied to much larger problems.
>
> We reply to the question of whether the types of problems in our experiments “can be really considered a proof-of concept”. First, we have updated the paper with new experimental results in Figure 6d on page 9 that shows the test of an 8x8-trained IPN policy on a 200x200 test mesh. We see that the trained policy still outperforms the ZZ and TrueError baselines on circle initial conditions, despite the 625x increase in the number of elements. Average times per action by the baselines and the RL policy were on the same order of magnitude. We wish to clarify that 200x200 refers to the number of elements, whereas there are actually 200x200x3x3=360,000 solution nodes. More generally, as explained in Section 5.1, our approach is to train on small representative features with known solutions and deploy on new problems without needing an analytic solution at test time. Generalization experiments in Section 6.3 show this can be done with good results for increasing refinement budget, larger initial mesh size, and different initial conditions. We agree that showing results on different boundary conditions and varying other parameters of the problem can strengthen this paper. However, being the first attempt to demonstrate deep RL on AMR, we had to select only a subset of the multitude of experiments to meet the constraints of a 9-page conference paper. In the end, we chose to fulfill two basic criteria: 1) explore the landscape of possible neural architectures; 2) conduct experiments on diverse types of problems (static, advection, Burgers) with solution features that may arise locally in larger problems. Note that the experimental sections of this paper already fill nearly 4.5 out of the 9 allotted pages of this conference paper.

---

> > ### Comment · Reviewer_Gif3 · 2021-11-28
> > **Response to rebuttal**
> >
> > I thank the authors for taking the time to respond to my comments. There have been some improvements in the paper but I think there are still points that remain unclear.
> > I agree with authors that RL enables "the integration of h-, p-, and r-refinement into a unified strategy, which does not currently exist via traditional methods that require patching a set of heuristics." and this is actually one of the reasons for my rating.
> > However, while the additional experimental in Figure 6d on page 9 is a nice addition, I don' think this is a major example showing the true advantage of RL for AMR.
> > Using RL techniques in the PDE community for estimation/control is justified in a sense that online control using model-based approaches, e.g. adjoint-based analysis, is computationally so expensive that make them intractable. On the other hand, data-driven approaches should be employed for online applications and that could include RL. For tasks such as mesh generation, however, since this is by nature an "offline" task, out-of-the-box approaches like RL can be justified only if they outperform traditional methods in complicated scenarios. If authors had shown AMR based on RL can be superior to traditional methods for, say, compressible airflow around airfoil, that would make a strong case to make the rating even higher. But all examples (both PDEs and geometries) are not challenging enough to really demonstrate that RL is a doing something not tractable by traditional methods. I have to emphasize that reviewer thinks the paper is well-written and the idea is nice but without challenging examples the claims are still debatable.
> > Also, some of my concerns regarding such as "mesh quality check" are not directly addressed.
> >
> > Regarding "... adjoint-based error estimators, those approaches are generally only practical for steady or quasi-steady problems" I'm not sure if that's entirely true (this is not really a fundamental restriction). See for example "Li and Petzold. Adjoint sensitivity analysis for time-dependent partial differential equations with adaptive mesh refinement. JCP 198.1 (2004): 310-325.". Many studies using adjoint-based AMR report meaningful improvements that are also scalable. Regarding computational burdens, I agree with authors that such AMR methods do suffer from complications in the numerical analysis, but there have been many advancements in development of differentiable solvers that can take care of such complexities. For transient problems with large time integration, checkpointing may be required but I'm sure RL would also become more challenging for such tasks. Maybe a comparison of the two methods would clarify all these points.
> >
> > Regarding "questions about the surrogate reward,.."; thanks for clarification. I agree with your response.
> >
> > Regarding"mathematical notation" I wasn't expecting a full terminology; only that point regarding "definition of test function" was necessary, which authors addressed.

---

> > > ### Author Response · Authors · 2021-11-29
> > > **We believe we have shown that RL is good enough at generalization on a diverse set of test problems that it is worthwhile for further research.**
> > >
> > > We believe our contributions should be evaluated on this basis: does the paper show that RL is good enough at generalization on a diverse set of test problems in comparison to an oracle baseline using the true error, that it is worthwhile for researchers at the intersection of machine learning and computational science and engineering to pursue this direction further, reveal strengths and weaknesses, and build upon this work to improve a model-free sample-based RL approach?
> > >
> > > We strongly believe the answer is "yes."
> > >
> > > Reviewer Gif3 says that “out-of-the-box approaches like RL can be justified only if they outperform traditional methods in complicated scenarios.” We ask: justified in what context? Justified for use in critical real-world applications? Justified for publication at a machine learning conference? These are two different goals, and the standards are different for each. Nowhere in this paper do we claim that this approach or any other RL-based approach is ready for use in real-world applications. Our claim is that RL has shown enough effectiveness in comparison to a strong oracle baseline that it is worth publication at a machine learning venue for more researchers in machine learning and computational science and engineering to pursue this research direction.
> > >
> > > As a first paper on RL for AMR, we made the following contributions: 1) a clear MDP formulation of AMR; 2) three effective policy architectures for variable-sized state/action spaces; 3) showing experimental results on diverse problems that AMR policies can be trained by RL on small representative problems, and either outperform or be competitive with the strong oracle TrueError baseline even when deployed in test problems that the policy has never seen before. In particular, we have shown the following advantage of RL:
> > > 1. policies trained on static function estimation can generalize to the advection equation
> > > 2. policies trained on small meshes can generalize to test meshes that are orders of magnitude larger
> > > 3. policies trained with small refinement budgets and short episode lengths can generalize to larger refinement budgets and longer episode lengths
> > > 4. for problems where exact solutions are difficult to compute in training, policies can be trained using a surrogate reward and still remain highly effective at test time.
> > > 5. policies trained on a single initial condition for Burger’s equation can be fine-tuned on random initial conditions and outperform the oracle TrueError baseline.
> > >
> > > We believe these are sufficient results to show that RL for AMR is worth publication at a machine learning venue.

---

### Official Review · Reviewer_sdD7 · 2021-11-08

**Correctness:** 3
**Technical Novelty And Significance:** 3
**Empirical Novelty And Significance:** 3
**Recommendation:** 6
**Confidence:** 3

**Main Review:**

Pros:
- A MDP formulation for mesh refinement
- Different policy architectures to capture mesh geometry
- Experiment on different settings: time-dependent, state and action size changes, and generalization to unseen mesh sizes and varied budget.

Cons:

- Unclear MDP formulation
- Policy network architectures are not novel.


In overall, the paper pursues an interesting direction. Dynamical mesh refinement is an important feature in numerical simulation of PDEs. It makes sense to formulate the process of dynamical mesh refinement as a MDP where state and action are well defined. It's also a complex problem if the dimensionality of state and action is allowed to vary over time.

1. At first, the description in section 3 can cover a bit more background of AMR so that the readers can follow and get ideas on how the formulation in 3.2 works.

2. Regarding the formulation in 3.2, as the authors define a MDP for a new sequential decision making problem, this formulation can be more complete, if it includes detailed definition of transition. It's hard to understand how transition is computed and defines the dynamics based on only short description. The same issue for the reward defection is hard to understand how those values are computed. It be helpful if the authors consider providing more description of AMR background.

3. Three policy architectures are good. However I could not find where in the paper the policy is designed to capture time-dependency and the change of state and action sizes.

4. The experiments are interesting which can show the benefit of the proposed approach. I have only one point that is hard to follow. Why the TrueError is worse than other methods. What is the implication of this baseline? Does it mean the metrics used to define rewards is not optimal? If so, can the formulation be different, e.g. taking into the downstream task's reward instead of direct mesh refinement's error?


**Summary Of The Paper:**

This paper proposes an application of reinforcement learning for adaptive mesh refinement in large-scale finite element simulations of complex physical systems. The authors suggest to formulate the mesh refinement problem as a MDP and propose different policy architectures for scalable application of reinforcement learning. Experiment results demonstrate that the proposed RL approaches outperform existing baselines, and can generalize well to situations of different finement budgets and larger meshes.

**Summary Of The Review:**

See above
===========================
After the rebuttal: Thanks the authors for the detailed response.
Though I found the proposed idea of using RL for AMR is interesting, the technical contribution is still limited.
Therefore I like to keep my original rating.

---

> ### Author Response · Authors · 2021-11-14
> **We address Reviewer sdD7's questions on 1) MDP transition dynamics, 2) how our method captures time dependency, 3) how our policies handle changing state-action sizes, 4) performance of TrueError baseline, 5) reward definition**
>
> We appreciate Reviewer sdD7's time spent on the review and the detailed questions, which we address here:
>
> 1. We can expand on the background of FEM and AMR in our revision as much as possible within page constraints (possibly in the appendices). One reason for providing only a high-level description is that the reinforcement learning algorithm only observes the current estimated solution (and possibly gradients) and the elements’ depth, without using any information about how exactly the FEM calculations arrived at the solution. For readers whose background lies more in RL than in FEM, the refinement process of 1) choosing an element, 2) splitting it into finer elements, 3) computing the solution on the new mesh can be intuitively understood at the schematic level shown in Figure 1 and the animations at our submitted hyperlink https://sites.google.com/view/iclr2022-amr
> 2. We defined the high-level points of the MDP transition in bullet points 1-3 in Section 3.2, and we will include more details about stepping a simulation forward in time (for time-dependent PDEs) and computing solutions on the new mesh. We chose to focus more on other aspects of the MDP, such as the definition of the variable-sized observation space and the reward, rather than the MDP transition because: 1) computing a solution on a mesh is a standard procedure in FEM; 2) our model-free RL approach learns from experiences of state-action-reward tuples and does not use any prior knowledge of the specific details of the transition dynamics.
> 3. Regarding “time dependency” and “change of state and action sizes”:
>     - 3a. Regarding how the model is designed to “capture time dependency”, the answer is: by virtue of reinforcement learning. The policy is trained to maximize the cumulative discounted reward over time. Concretely, for policy-gradient methods, this means that at any current state $s$, all future rewards collected by the agent at subsequent states $s’$ are used to update the model that determines the policy’s behavior at $s$. Hence, the policy is trained to act not merely greedily at $s$, but to account for the dependence of future rewards on its current action at $s$.
>     - 3b. Regarding how the model is designed to capture “the change of state and action sizes,” the answer is: by the very construction of the policy networks. Take the IPN for example in equation (3), the trainable function $f_{\theta}$ is independently applied to each element observation $o^i$, for $i =1,\dotsc,N$ where $N$ is the number of mesh elements and changes at each time step. Since the policy is a softmax over all $f_{\theta}(o^i)$, $i =1,\dotsc,N$, as shown in equation (3), it accommodates the varying size of state and action sets. The case for Hypernetwork is similar to IPN. In the case of the Graphnet policy, the trainable functions are edge representation network $\phi^e$ and node representation network $\phi^v$. As described in the second paragraph in Section 4.3, these two networks are applied to each edge and node individually (in parallel), which accommodates the varying number of edges and nodes as the mesh size increases.
> 4. The TrueError baseline refines the element with the largest error, but this does not necessarily result in the largest reduction of error, which is the actual optimization objective. This is the reason that TrueError performs below the GreedyOptimal baseline, as shown in Figure 3(a,b). Nonetheless, TrueError serves as an approximate upper bound on the performance of an instantaneous error estimator, as shown by the fact that TrueError outperforms the ZZ baseline in almost all cases (note: refining the element with highest true error may not lead to the largest error reduction, so it is possible but rare that another error-estimation-based method outperforms TrueError simply due to random chance). We emphasize again that TrueError cannot be deployed on systems without known solutions and GreedyOptimal is intractable for real applications, unlike our RL method.
>     - 4a. Regarding whether the “metrics used to define rewards is not optimal”. We point out that “optimality” is defined in terms of maximizing cumulative rewards, not the other way around. So, the question really is asking whether the reward function we defined in equation (1) makes sense. We believe it does: maximizing the sum of the reward in equation (1) over time is exactly equivalent to maximizing the reduction of error (subject to a refinement budget constraint), which is the true objective in AMR.

---

### Author Response · Authors · 2021-11-14
**We have uploaded a revision of this paper**

We have uploaded a revision of this paper, with the following changes:
1. We have added new experimental results in Figure 6d on page 9 that shows the test of an 8x8-trained IPN policy on a 200x200 test mesh. We see that the trained policy still outperforms the ZZ and TrueError baselines on circle initial conditions, despite the 625x increase in the number of elements. Average times per action by the baselines and the RL policy were on the same order of magnitude.
2. We have removed a sentence in Section 3.2 that said "such variation in state-action spaces does not occur in any existing RL application".  This does not impact the overall significance of this work. Our intended meaning is already expressed more accurately by a sentence in Section 2, which says RL has been "typically applied to environments with fixed-size observation and small bounded action spaces in almost all benchmark problems."

---

### Comment · Area_Chair_TknN · 2021-11-29
**Any final thoughts?**

Dear reviewers,

This is a follow-up on the private messages I sent through OpenReview. If you haven't already done so, please read the authors' response, and engage in the final discussions with them. At the very least, acknowledge their responses, and indicate whether they adequately addressed your concerns or not.

Today (November 29th) is the end of the discussion period.

Thank you, Area Chair

---

### Decision · Program_Chairs · 2022-01-20

**Decision:**

Reject

**Comment:**

This work formulates the Adaptive Mesh Refinement (AMR) problem used in solving Finite Element Method (FEM) as an MDP, and suggests an RL-based solution for it. Most reviewers agree that this is a novel problem and the solution is promising. There are, however, several issues raised by our reviewers, who have expertise ranging from ML to computational methods to solve PDEs. Some of the concerns are:

- As this is not a theoretical work, the burden of proof is on the empirical evaluations. Some reviewers found the experiments very small and not convincing enough.
- The paper does not compare with the state of the art AMR methods.
- The detail of how the problem is formulated as an MDP can be improved.

Given that four out of five reviewers are on the negative side, unfortunately I cannot recommend acceptance of this paper in its current form. Nevertheless, I believe this is a promising application of RL. I'd like to encourage the authors to consider the reviews in order to improve their work, and resubmit it to another venue.